# Minimum Description Length Control

**Ted Moskovitz[1,*], Calvin Kao[2,3], Maneesh Sahani[1], Matthew M. Botvinick[1,2]**

1. Gatsby Unit, University College London
2. DeepMind
3. Facebook Reality Labs
*Correspondence: ted@gatsby.ucl.ac.uk

## Abstract

We propose a novel framework for multitask reinforcement learning based on the *minimum description length* (MDL) principle. In this approach, which we term *MDL-control* (MDL-C), the agent learns the common structure among the tasks with which it is faced and then distills it into a simpler representation which facilitates faster convergence and generalization to new tasks. In doing so, MDL-C naturally balances adaptation to each task with epistemic uncertainty about the task distribution. We motivate MDL-C via formal connections between the MDL principle and Bayesian inference, derive theoretical performance guarantees, and demonstrate MDL-C's empirical effectiveness on both discrete and high-dimensional continuous control tasks.

## 1 Introduction

In order to learn efficiently in a complex world with multiple rapidly changing objectives, both animals and machines must leverage past experience. This is a challenging task, as processing and storing all relevant information is computationally infeasible. How can an intelligent agent address this problem? We hypothesize that one route may lie in the *dual process theory* of cognition, a longstanding framework in cognitive psychology introduced by William James (James, 1890) which lies at the heart of many dichotomies in both cognitive science and machine learning. Examples include goal-directed versus habitual behavior (Graybiel, 2008), model-based versus model-free reinforcement learning (Daw et al., 2011; Sutton and Barto, 2018), and "System 1" versus "System 2" thinking (Kahneman, 2011). In each of these paradigms, a complex, "control" process trades off with a simple, "default" process to guide actions. Why has this been such a successful and enduring conceptual motif? Our hypothesis is that default processes often serve to distill common structure from the tasks consistently faced by animals and agents, facilitating generalization and rapid learning on new objectives. For example, drivers can automatically traverse commonly traveled roads en route to new destinations, and chefs quickly learn new dishes on the back of well-honed fundamental techniques. Importantly, even intricate tasks can become automatic, if repeated often enough (e.g., the combination of fine motor commands required to swing a tennis racket): the default process must be sufficiently expressive to learn common behaviors, regardless of their complexity. In reality, most processes likely lie on a continuum between simplicity and complexity.

In reinforcement learning (RL; Sutton and Barto, 2018), improving sample efficiency on new tasks is crucial to the developement of general agents which can learn effectively in the real world (Botvinick et al., 2015; Kirk et al., 2021). Intriguingly, one family of approaches which have shown promise in this regard are *regularized policy optimization* algorithms, in which a goal-specific control *policy* is paired with a simple yet general default *policy* to facilitate learning across multiple tasks (Teh et al., 2017; Galashov et al., 2019; Goyal et al., 2020; 2019; Moskovitz et al., 2022a). One difficulty in algorithm design, however, is how much or how little to constrain the default policy, and in what way. An overly simple default policy will fail to identify and exploit commonalities among tasks, while a complex model may overfit to a single task and fail to generalize. Most approaches manually specify an asymmetry between the control and default policies, such as hiding input information (Galashov et al., 2019) or constraining the model class (Lai and Gershman, 2021). Ideally, we'd like an adaptive approach that learns the appropriate degree of complexity via experience.

The *minimum description length principle* (MDL; Rissanen, 1978), which in general holds that one should prefer the simplest model that accurately fits the data, offers a guiding framework for algorithm design that does just that, enabling the default policy to optimally trade off between adapting to information from new tasks and maintaining simplicity. Inspired by dual process theory and the MDL principle, we propose *MDL-control* (MDL-C, pronounced "middle-cee"), a principled RPO framework for multitask RL. In Section 2, we formally introduce multitask RL and describe RPO approaches within this setting. In Section 3, we describe MDL and the variational coding framework, from which we extract MDL-C and derive its formal performance characteristics. In Section 5, we demonstrate its empirical effectiveness in both discrete and continuous control settings. Finally, we discuss related ideas from the the literature (Section 6) and conclude (Section 7).

## 2 REINFORCEMENT LEARNING PRELIMINARIES

**The single-task setting** We model a task as a *Markov decision process* (MDP; Puterman, 2010) $M = (\mathcal{S}, \mathcal{A}, \mathsf{P}, r, \gamma, \rho)$, where $\mathcal{S}, \mathcal{A}$ are state and action spaces, respectively, $\mathsf{P} : \mathcal{S} \times \mathcal{A} \to \mathcal{P}(\mathcal{S})$ is the state transition distribution, $r : \mathcal{S} \times \mathcal{A} \to [0, 1]$ is a reward function, $\gamma \in [0, 1)$ is a discount factor, and $\rho \in \mathcal{P}(\mathcal{S})$ is the starting state distribution. $\mathcal{P}(\cdot)$ is the space of probability distributions defined over a given space. The agent takes actions using a *policy* $\pi : \mathcal{S} \to \mathcal{P}(\mathcal{A})$. In large or continuous domains, the policy is often parameterized: $\pi \to \pi_\theta$, $\theta \in \Theta$, where $\Theta \subseteq \mathbb{R}^d$ represents a particular model class with $d$ parameters. In conjunction with the transition dynamics, the policy induces a distribution over trajectories $\tau = (s_h, a_h)_{h=0}^{\infty}$, $\mathsf{P}^{\pi_\theta}(\tau)$. In a single task, the agent seeks to maximize its *value* $V^{\pi_\theta} = \mathbb{E}_{\tau \sim \mathsf{P}^{\pi_\theta}} R(\tau)$, where $R(\tau) := \sum_{h \geq 0} \gamma^h r(s_h, a_h)$ is called the *return*. We denote by $d_\rho^\pi$ the state-occupancy distribution induced by policy $\pi$ with starting state distribution $\rho$: $d_\rho^\pi(s) = \mathbb{E}_\rho (1 - \gamma) \sum_{h \geq 0} \gamma^h \Pr(s_h = s | s_0)$.

**Multiple tasks** There are a number of frameworks for multitask RL in the literature (Yu et al., 2019; Zahavy et al., 2021; Finn et al., 2017; Brunskill and Li, 2013). For a more detailed discussion, see Appendix Section B. In this paper, we focus primarily on what we term the *sequential* and *parallel* task settings. The objective in each case is to maximize *average reward across tasks*, equivalent to minimizing cumulative regret over the agent's 'lifetime.' More specifically, we assume a (possibly infinite) set of tasks (MDPs) $\mathcal{M} = \{M\}$ presented to the agent by sampling from some task distribution $\mathsf{P}_\mathcal{M} \in \mathcal{P}(\mathcal{M})$. In the *sequential* task setting (Moskovitz et al., 2022a; Pacchiano et al., 2022), tasks (MDPs) are sampled one at a time from $\mathsf{P}_\mathcal{M}$, with the agent training on each until convergence. In the parallel task training (Yu et al., 2019), a new MDP is sampled from $\mathsf{P}_\mathcal{M}$ at the start of every episode and is associated with a particular input feature $g \in \mathcal{G}$ that indicates to the agent which task has been sampled.

**Regularized Policy Optimization** One common approach which improves performance is *regularized policy optimization* (RPO; Schulman et al., 2017; 2018; Levine, 2018; Agarwal et al., 2020; Pacchiano et al., 2020; Tirumala et al., 2020; Abdolmaleki et al., 2018). In RPO, a convex regularization term $\Omega(\theta)$ is added to the objective: $\mathcal{J}_\lambda^{\mathrm{RPO}}(\theta) = V^{\pi_\theta} - \lambda \Omega(\theta)$. In the single-task setting, the regularization term is often used to approximate trust region (Schulman et al., 2015), proximal point (Schulman et al., 2017), or natural gradient (Kakade, 2002; Pacchiano et al., 2020; Moskovitz et al., 2020) optimization, or to prevent premature convergence to local maxima (Haarnoja et al., 2018; Lee et al., 2018). In multitask settings, the regularization term for RPO typically takes the form of a divergence measure penalizing the policy responsible for taking actions $\pi_\theta$, which we'll refer to as the *control policy*, for deviating from some *default policy* $\pi_w$, which is intended to encode generally useful behavior for some family of tasks (Teh et al., 2017; Galashov et al., 2019; Goyal et al., 2019; 2020; Moskovitz et al., 2022a). By capturing behavior which is on average useful across tasks, $\pi_w$ can provide a form of beneficial supervision to $\pi_\theta$ when obtaining reward is challenging, either because $\pi_\theta$ has been insufficiently trained or rewards are sparse.

## 3 THE MINIMUM DESCRIPTION LENGTH PRINCIPLE

**General principle** Storing all environment interactions across multiple tasks is computationally infeasible, so multitask RPO algorithms offer a compressed representation in the form of a default policy. However, the type of information which is compressed (and that which is lost) is often

hard-coded *a priori*. Preferably, we'd like an approach which can distill structural regularities among tasks without needing to know what they are beforehand. The *minimum description length* (MDL) framework offers a principled approach to this problem. So-called "ideal" MDL seeks to find the shortest solution written in a general-purpose programming language[1] which accurately reproduces the data—an idea rooted in the concept of Kolmogorov complexity (Li and Vitnyi, 2008). Given the known impossibility of computing Kolmogorov complexity for all but the simplest cases, a more practical MDL approach instead prescribes selecting the hypothesis $H^\star$ from some hypothesis class $\mathcal{H}$ which minimizes the two-part code $H^\star = \operatorname{argmin}_{H \in \mathcal{H}} L(\mathcal{D}|H) + L(H)$, where $L(\mathcal{D}|H)$ is the number of bits required to encode the data given the hypothesis and $L(H)$ is the number of bits needed to encode the hypothesis itself. There are a variety of so-called *universal* coding schemes which can be used to model these quantities.

**Variational code** One popular encoding scheme is the variational code (Blier and Ollivier, 2018; Hinton and Van Camp, 1993; Honkela and Valpola, 2004):

$$L_\nu^{\mathrm{var}}(\mathcal{D}) = \underbrace{\mathbb{E}_{\theta \sim \nu}\left[-\log p_\theta(\mathcal{D})\right]}_{L^{\mathrm{var}}(\mathcal{D}|H)} + \underbrace{\mathsf{KL}[\nu(\cdot), p(\cdot)]}_{L^{\mathrm{var}}(H)} \tag{3.1}$$

where the hypothesis class is of a set of parametric models $\mathcal{H} = \{p_\theta(\mathcal{D}) : \theta \in \Theta\}$. The model parameters are random variables with prior distribution $p(\theta)$ and $\nu(\theta)$ is any distribution over $\Theta$. Minimizing $L_\nu^{\mathrm{var}}(\mathcal{D})$ with respect to $\nu$ is equivalent to performing variational inference, maximizing a lower-bound to the data log-likelihood $\log p(\mathcal{D}) = \log \int p(\theta) p_\theta(\mathcal{D}) d\theta \geq -L_\nu^{\mathrm{var}}(\mathcal{D})$. Roughly speaking, MDL encourages the choice of "simple" models when limited data are available (Grunwald, 2004). In the variational coding scheme, simplicity is enforced via the choice of prior.

**Sparsity-inducing priors and variational dropout** Sparsity-inducing priors can be used to improve the compression rate within the variational coding scheme, as they encourage the model to prune out parameters that do not contribute to reducing $L^{\mathrm{var}}(\mathcal{D}|\theta)$. Many sparsity-inducing priors belong to the family of scale mixtures of normal distributions (Andrews and Mallows, 1974): $z \sim p(z)$, $\theta \sim p(\theta|z) = \mathcal{N}(w; 0, z^2)$ where $p(z)$ defines a distribution over the variance $z^2$. Common choices of $p(z)$ include the Jeffreys prior $p(z) \propto |z|^{-1}$ (Jeffreys, 1946), the inverse-Gamma distribution, and the half-Cauchy distribution (Polson and Scott, 2012; Gelman, 2006). Such priors have deep connections to MDL theory. For example, the Jeffreys prior in conjunction with an exponential family likelihood is asymptotically identical to the *normalized maximum likelihood* estimator, perhaps the most fundamental 'MDL' estimator (Grünwald and Roos, 2019). *Variational dropout* (VDO) is an effective algorithm for minimizing Equation (3.1) for these sparsity-inducing priors (Louizos et al., 2017; Kingma et al., 2015; Molchanov et al., 2017). Briefly, this involves choosing an approximate posterior distribution with the form

$$p(w, z|\mathcal{D}) \approx \nu(w, z) = \mathcal{N}(z; \mu_z, \alpha\sigma_z^2)\mathcal{N}(w; z\mu, z^2\sigma^2 I_d) \tag{3.2}$$

and optimizing Equation (3.1) via stochastic gradient descent on the variational parameters given by $\{\alpha, \mu_z, \sigma_z^2, \mu, \sigma^2\}$. As its name suggests—and importantly for its ease of application to large models—VDO can be implemented as a form of dropout (Srivastava et al., 2014) by reparameterizing the noise on the weights as activation noise (Kingma et al., 2015). Application of VDO to Bayesian neural networks has achieved impressive compression rates, sparsifying deep neural networks while maintaining prediction performance on supervised learning problems (Molchanov et al., 2017; Louizos et al., 2017). Equipped with a powerful approach for MDL-grounded posterior inference, we can now integrate these ideas with multitask RPO.

## 4 MINIMUM DESCRIPTION LENGTH CONTROL

As part of its underlying philosophy, the MDL principle holds that 1) *learning* is the process of discovering regularity in data, and 2) any regularity in the data can be used to *compress* it (Grunwald, 2004). Applying this perspective to RL is non-obvious—from the agent's perspective, what 'data' is it trying to compress? Our hypothesis, which forms the basis for the framework

---

[1]The *invariance theorem* (Kolmogorov, 1965) ensures that, given a sufficiently long sequence, Kolmogorov complexity is invariant to the choice of general-purpose language.

---

**Algorithm 1:** MDL-C for Sequential Multitask Learning with Persistent Replay

---

1: Require: task distribution $\mathsf{P}_{\mathcal{M}}$, policy class $\Theta$, non-increasing coefficients $\{\eta_k\}_{k=1}^K$
2: Initialize: default policy distribution $\nu_1 \in \mathsf{N} \subseteq \mathcal{P}(\Theta)$, default policy dataset $\mathcal{D}_0 \leftarrow \emptyset$
3: **for** tasks $k = 1, 2, \ldots, K$ **do**
4:    Sample a task $M_k = (\mathcal{S}, \mathcal{A}, \mathsf{P}_k, r_k, \gamma_k, \rho_k) \sim \mathcal{P}_{\mathcal{M}}(\cdot)$
5:    Optimize control policy:

$$\theta_k \leftarrow \underset{\theta \in \Theta}{\arg\max} \, V_{M_k}^{\pi} - \alpha \mathbb{E}_{s \sim d_{\rho_k}^{\pi}} \mathbb{E}_{w \sim \nu_k} \mathsf{KL}[\pi_w(\cdot|s), \pi_\theta(\cdot|s)] \qquad (4.2)$$

6:    Add data to default policy replay ($M = |\mathcal{S}|$ for finite/small state spaces):

$$\mathcal{D}_k \leftarrow \mathcal{D}_{k-1} \cup \{(s_m, \hat{\pi}_{\theta_k}(s_m))\}_{m=1}^M. \qquad (4.3)$$

7:    Update default policy distribution:

$$\nu_{k+1} \leftarrow \underset{\nu \in \mathsf{N}}{\arg\min} \, \frac{1}{\eta_{k-1}} \mathsf{KL}[\nu(\cdot), p(\cdot)] + \sum_{i=1}^k \sum_{m=1}^M \mathbb{E}_{w \sim \nu} \mathsf{KL}[\hat{\pi}_{\theta_i}^\star(\cdot|s_m), \pi_w(\cdot|s_m)] \qquad (4.4)$$

8: **end for**

---

we propose in this paper, is that an agent faced with a set of tasks in the world should seek to elucidate structural regularity from the environment interactions generated by the optimal policies for the tasks. This makes intuitive sense: the agent ought to compress information which indicates how to correctly perform the tasks with which it is faced. That is, we propose that the *data* in multitask RL are the state-action interactions generated by the optimal policies for a set of tasks: $\mathcal{D} = \{\mathcal{D}_M\}_{M \in \mathcal{M}} = \{(s, a) \, : \, \forall s \in \mathcal{S}, a \sim \pi_M^\star(\cdot|s)\}_{M \in \mathcal{M}}$ This interpretation is in line with work suggesting that a useful operational definition of 'task' can be derived directly from the set of optimal (or near-optimal) policies it induces (Abel et al., 2021). It also suggests a natural mapping to the multitask RPO framework. In this view, the control policy is responsible for learning and the default policy for compression: by converging to the optimal policy for a given task, the control policy "discovers" regularity which is then distilled into a low-complexity representation by the default policy. In our approach, the default policy is encouraged to learn a compressed representation not by artificially constraining the network architecture or via hand-designed information asymmetry, but rather through a prior distribution $p(w)$ over its parameters which biases a variational posterior $\nu(w)$ towards simplicity. The default policy is therefore trained to minimize the variational code:

$$\begin{aligned} \underset{\nu \in \mathsf{N}}{\arg\min} \, \mathbb{E}_{\substack{s, a \sim \mathcal{D} \\ w \sim \nu}} &- \log \pi_w(a|s) + \mathsf{KL}[\nu(\cdot), p(\cdot)] \\ &= \underset{\nu \in \mathsf{N}}{\arg\min} \, \mathbb{E}_{M \sim \mathsf{P}_{\mathcal{M}}} \mathbb{E}_{\substack{s \sim d_M^{\pi_M^\star} \\ w \sim \nu_\phi}} \mathsf{KL}[\pi_M^\star(\cdot|s), \pi_w(\cdot|s)] + \mathsf{KL}[\nu(\cdot), p(\cdot)], \end{aligned} \qquad (4.1)$$

where $\mathsf{N}$ is the distribution family for the posterior. This suggests the approach presented in Algorithm 1, in which for each task $M_k$, the control policy $\pi_\theta$ is trained to approximate the optimal policy $\pi_k^\star$ via RPO, and the result is compressed into a new default policy distribution $\nu_{k+1}$. We now further motivate sparsity-inducing priors for the default policy in multitask settings, derive formal performance guarantees for MDL-C, and demonstrate its empirical effectiveness.

## 4.1 MOTIVATING THE CHOICE OF SPARSITY-INDUCING PRIORS

In Section 3, compression (via pruning extraneous parameters) is the primary motivation for using sparsity-inducing priors that belong to the family of scaled-mixtures of normal distributions. Intuitively, placing a distribution over the default parameters reflects the agent's *epistemic uncertainty* about the task distribution—when few tasks have been sampled, a sparse prior prevents the default policy from overfitting to spurious correlations in the limited data that the agent has collected. Here, we make this motivation more precise, describing an example generative model of optimal policy parameters which provides a principled interpretation for prior choice $p(z)$ in multitask RL.

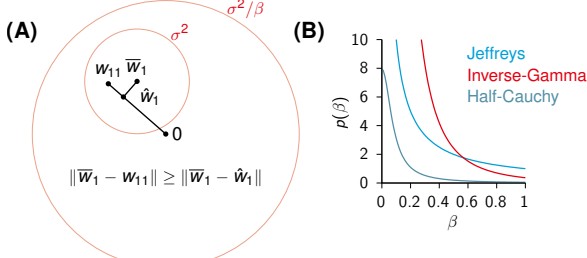

Figure 4.1: **(A)** Illustration of a generative model of optimal policy parameters. $\hat{w}_1 = (1 - \beta)w_{11}$ shrinks towards the origin, growing closer to $\overline{w}_1$ than $w_{11}$. **(B)** Sparsity-inducing priors over $\beta$.

**Generative model of optimal policy parameters**   Consider a set of tasks $\mathcal{M} = \{M_{ik}\}_{i=1, k=1}^{I, K_i}$ that are clustered into $I$ groups, such that the MDPs in each group are more similar to one another than to members of other groups. As an example, the overall family $\mathcal{M}$ could be all sports, while clusters $\mathcal{M}_i \subseteq \mathcal{M}$ could consist of, say, ball sports or endurance competitions. To make this precise, we assume that the optimal policies of every MDP belong to a parametric family $\Pi = \{\pi_w(\cdot|s) : w \in \mathbb{R}^d, \forall s \in \mathcal{S}\}$ (e.g., softmax policies with parameters $w$), and that the optimal policies for each group are randomly distributed within parameter space. In particular, we assume that the parameters of the optimal policies of $\mathcal{M}$ have the following generative model:

$$\overline{w}_i|\beta, \sigma^2 \sim \mathcal{N}\left(\overline{w}_m; 0, (1-\beta)\beta^{-1}\sigma^2 I_d\right), \quad w_{ik}|\overline{w}_i, \sigma^2 \sim \mathcal{N}\left(w_{ik}; \overline{w}_i, \sigma^2 I_d\right).$$

where $I_d$ is the $d-$dimensional identity matrix. If we marginalize out $\overline{w}_i$, we get the marginal distribution $p(w_{ik}|\beta, \sigma^2) = \mathcal{N}(w_{ik}; 0, \sigma^2\beta^{-1}I_d)$. We can then visualize the parameter distribution of the optimal policies for $\mathcal{M}$ as a $d$-dimensional Gaussian within which lie clusters of optimal policies for related tasks which are themselves normally distributed (see Fig. 4.1A for $d = 2$).

**Interpretation of $\beta$**   The parameter $\beta \in (0, 1]$ can be interpreted as encoding the squared distance between optimal policy parameters within a group divided by the squared distance between optimal policies in $\mathcal{M}$. Intuitively, $\beta$ determines how much information one gains about the optimal parameters of a task in a group, given knowledge about the optimal parameters of another task in the same group. To see this, we compute our posterior belief about the value $\overline{w}_i$ given observation of $w_{ik}$: $p(\overline{w}_i|w_{ik}, \beta, \sigma^2) = \mathcal{N}\left(\overline{w}_i; (1-\beta)w_{ik}, (1-\beta)\sigma^2 I_d\right)$. When $\beta = 1$ (inner circle in Figure 4.1A has the same radius as the outer circle), our posterior mean estimate of $\overline{w}_i$ is simply 0, suggesting we have learned nothing new about the mean of the optimal parameters in group $i$, by observing $w_{ik}$. In the other extreme when $\beta \to 0$, the posterior mean approaches the maximum-likelihood estimator $w_{ik}$, suggesting that observation of $w_{ik}$ provides maximal information about the optimal parameters in group $i$. Any $\beta$ in between the two extremes results in an estimator that "shrinks" $w_{ik}$ towards 0. The value of $\beta$ thus has important implications for multitask learning. Suppose an RL agent learns the optimal parameters $w_{11}$ (task 1, group 1), and proceeds to learn task 2 in group 1. The value of $\beta$ determines whether $w_{11}$ can be used to inform the agent's learning of $w_{21}$. In this way, $\beta$ determines the effective degree of epistemic uncertainty the agent has about the task distribution.

**Choice of $p(\beta)$ and connection to $p(z)$**   Given its importance, it's natural to ask what value $\beta$ should take. Instead of treating $\beta$ as a parameter, we can choose a prior $p(\beta)$ and perform Bayesian inference. Ideally, $p(\beta)$ should (i) encode our prior belief about the extent to which the optimal parameters cluster into groups and (ii) result in a posterior mean estimator $\hat{w}^{(p(\beta))}(x) = 1 - \mathbb{E}[\beta|x]\,x$ that is close to $\overline{w}$ for $x|\overline{w} \sim \mathcal{N}(x; \overline{w}, \sigma^2)$. This condition encourages the expected default policy (under the posterior $\nu$; Equation (4.1)) to be close to optimal policies in the same MDP group (centered at $\overline{w}$). One prior choice that satisfies both conditions is $p(\beta) \propto \beta^{-1}$. It places high probability for small $\beta$ and low probability for high $\beta$, thus encoding the prior belief that the optimal task parameters are clustered (see Figure 4.1B; blue). It is instructive to compare $p(\beta) \propto \beta^{-1}$ with two extreme choices of $p(\beta)$. When $p(\beta) = \delta(\beta - 1)$, $p(z) = \delta(\sigma)$ and the marginal $p(w)$ is the often-used Gaussian prior over the parameters $w$ with fixed variance $\sigma^2$. This corresponds to the prior belief that knowing $w_{i1}$ provides no information about $w_{i2}$. On the other hand, $p(\beta) = \delta(\beta)$ recovers a uniform prior over the parameters $w$ and reflects the prior belief that the MDP groups are infinitely far apart. In relation to (ii), one can show the $\hat{w}^{(p(\beta))}$ strictly dominates the maximum-likelihood estimator $\hat{w}^{(\text{ML})}(x) = x$ (Efron

and Morris, 1973; Section D), for $p(\beta) \propto \beta^{-1}$. This means $\mathrm{MSE}(\overline{w}, \hat{w}^{(p(\beta))}) \leq \mathrm{MSE}(\overline{w}, \hat{w}^{(\mathrm{ML})})$ for all $\overline{w}$, where $\mathrm{MSE}(\overline{w}, \hat{w}) = \mathbb{E}_{x \sim \mathcal{N}(x;\overline{w},\sigma^2)} \|\overline{w} - \hat{w}(x)\|^2$.

**Connection to $p(z)$ and application of VDO** Defining $z^2 = \sigma^2 \beta^{-1}$ and applying the change-of-variable formula to $p(\beta) \propto \beta^{-1}$ gives $p(z) \propto |z|^{-1}$ and thus the Normal-Jeffreys prior in Section 3. VDO (see Section 3) can then be applied to obtain an approximate posterior $\nu(w, z)$ which minimizes the variational code Equation (4.1). Similar correspondences may also be derived for the inverse-Gamma distribution and the half-Cauchy distribution (Figure 4.1B; Section D).

## 4.2 PERFORMANCE ANALYSIS

At a fundamental level, we'd like assurance (i) that MDL-C's default policy will be able to effectively distill the optimal policies for previously observed tasks, and (ii) that regularization using this default policy gives strong performance guarantees for the control policy on future tasks.

**Default policy performance** One way we can verify (i) is to obtain an upper bound on the average KL between default policies sampled from the default policy distribution and an optimal policy for a task sampled from the task distribution. This enables us to perform analysis using *online convex optimization* (OCO). In OCO, the learner observes a series of convex loss functions $\ell_k : \mathsf{N} \to \mathbb{R}$, $k = 1, \ldots, K$, where $\mathsf{N} \subseteq \mathbb{R}^d$ is a convex set. After each round, the learner produces an output $x_k \in \mathsf{N}$ for which it will then incur a loss $\ell_k(x_k)$ (Orabona, 2019). At round $k$, the learner is usually assumed to have knowledge of $\ell_1, \ldots, \ell_{k-1}$, but no other assumptions are made about the sequence of loss functions. The learner's goal is to minimize its average regret. For further background, see Section F. Crucially, the MDL-C learning procedure for the default policy distribution is equivalent to *follow the regularized leader* (FTRL), an OCO algorithm which enjoys sublinear regret. In each round of FTRL, the learner selects the solution $x \in \mathsf{N}$ according to the following general objective: $x_{k+1} = \mathrm{argmin}_{x \in \mathsf{N}} \psi_k(x) + \sum_{i=1}^{k-1} \ell_i(x)$, where $\psi : \mathsf{N} \to \mathbb{R}$ is a convex regularization function. Using standard results, this connection allows us to bound MDL-C's regret in learning the default policy distribution. All proofs are provided in Section G.

**Proposition 4.1** (Persistent Replay FTRL Regret). *Let tasks $M_k$ be independently drawn from $\mathsf{P}_\mathcal{M}$ at every round, and let them each be associated with a deterministic optimal policy $\pi_k^\star : \mathcal{S} \to \mathcal{A}$. We make the following mild assumptions: i) $\pi_w(a^\star|s) \geq \epsilon > 0 \ \forall s \in \mathcal{S}$, where $a^\star = \pi_k^\star(s)$ and $\epsilon$ is a constant. ii) $\min_\nu \mathsf{KL}[\nu(\cdot), p(\cdot)] = 0$ asymptotically as $\mathrm{Var}[\nu] \to \infty$. Then with $\eta_{k-1} = \log(1/\epsilon)\sqrt{k}$, Algorithm 1 guarantees*

$$\frac{1}{K}\sum_{k=1}^K \ell_k(\nu_k) - \frac{1}{K}\sum_{k=1}^K \ell_k(\bar{\nu}_K) \leq (\mathsf{KL}[\bar{\nu}_K, p] + 1)\frac{\log(1/\epsilon)}{\sqrt{K}}, \tag{4.5}$$

*where $\bar{\nu}_K = \mathrm{argmin}_{\nu \in \mathsf{N}} \sum_{k=1}^K \ell_k(\nu)$.*

**Control policy performance** Intuitively, this result shows that the average regret is upper-bounded by factors which depend on the divergence of the barycenter distribution from the prior and the "worst-case" prediction of the default policy. Importantly, the KL between the default policy distribution and the barycenter distribution goes to zero as $K \to \infty$. We can also now be assured of point (ii) above, in that this result can be used to obtain a sample-complexity bound for the *control* policy. Specifically, we can use Proposition G.1 to place an upper-bound on the total variation distance between default policies sampled from $\nu$ and the KL between the maximum likelihood solution and a sparsity-inducing prior $p$. This is useful, as it allows to translate low regret for the default policy into a sample complexity result for the control policy using Moskovitz et al. (2022a), Lemma 5.2.

**Proposition 4.2** (Control Policy Sample Complexity). *Under the setting described in Proposition G.1, denote by $T_k$ the number of iterations to reach $\epsilon$-error for $M_k$ in the sense that $\min_{t \leq T_k}\{V^{\pi_k^\star} - V^{(t)}\} \leq \epsilon$. whenever $t > T_k$. Further, denote the upper-bound in Eq. (G.1) by $G(K)$. In a finite MDP, from any initial $\theta^{(0)}$, and following gradient ascent, $\mathbb{E}_{M_k \sim \mathcal{P}_\mathcal{M}}[T_k]$ satisfies:*

$$\mathbb{E}_{M_k \sim \mathcal{P}_{\mathcal{M}_i}}[T_k] \geq \frac{80|\mathcal{A}|^2|\mathcal{S}|^2}{\epsilon^2(1-\gamma)^6}\mathbb{E}_{\substack{M_k \sim \mathcal{P}_{\mathcal{M}_i}\\ s \sim \mathrm{Unif}_\mathcal{S}}}\left[\kappa_\mathcal{A}^{\alpha_k}(s)\left\|\frac{d_\rho^{\pi_k^*}}{\mu}\right\|_\infty^2\right],$$

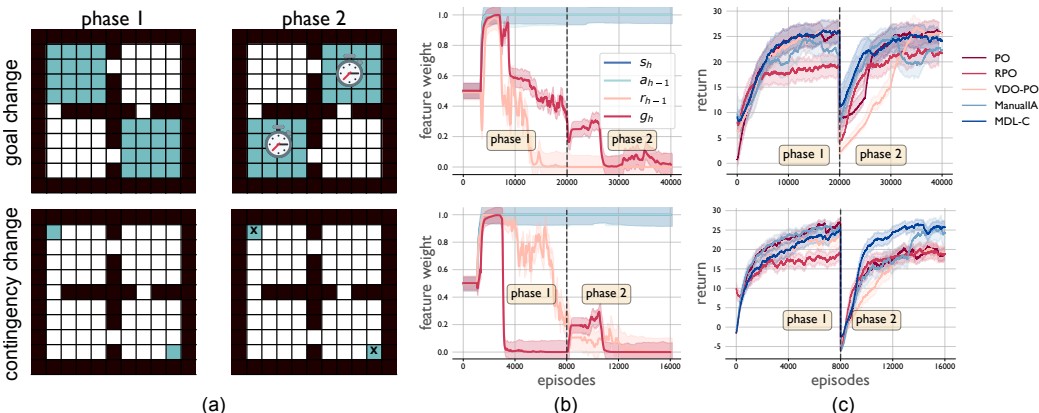

Figure 5.1: MDL-C rapidly adapts to new goal locations (top row) and rule changes (bottom row). All curves represent averages taken over 10 random seeds, with the shading indicating standard error.

where $\alpha_k(s) := d_{\text{TV}}(\pi_k^\star(\cdot|s), \hat{\pi}_0(\cdot|s)) \leq \sqrt{G(K)}$, $\kappa_{\mathcal{A}}^{\alpha_k}(s) = \frac{2|\mathcal{A}|(1-\alpha(s))}{2|\mathcal{A}|(1-\alpha(s))-1}$, and $\mu$ is a measure over $\mathcal{S}$ such that $\mu(s) > 0 \ \forall s \in \mathcal{S}$.

Intuitively, this means that when the average number of samples is sufficiently large, the control policy is guaranteed to have reached $\varepsilon$ error. Therefore, as the agent is trained on more tasks, the default policy distribution regret, upper-bounded by $G(K)$, decreases asymptotically to zero, and as the default policy regret decreases, the control policy will learn more rapidly, as $\text{poly}(G(K))$.

## 5 Experiments

We tested MDL-C applied to discrete and continuous control in both the sequential and parallel task settings. To quantify performance, in addition to measuring per-task reward, we also report the cumulative regret for each method in each experimental setting in Section I.1.

### 5.1 2D Navigation

We first test MDL-C in the classic FourRooms environment (Fig. 5.1a, (Sutton et al., 1999)). The baselines in this case are PO entropy-regularized policy optimization (PO), regularized policy optimization with no constraint on the default policy (RPO), an agent with VDO applied to the control policy and no default policy (VDO-PO), and ManualIA (Galashov et al., 2019) in which the goal feature is manually witheld from the default policy. Details can be found in Section H.

**Generalization Across Goals** In the first setting, we test MDL-C's ability to facilitate rapid learning on previously unseen goals. In the first phase of training, a single goal location is randomly sampled at the start of each episode, and may be placed anywhere in two of the four rooms in the environment (Fig. 5.1a, top left). In the second phase, the goal location is again randomly sampled at the start of each episode, but in this case, only in the rooms which were held out in the first phase. Additionally, the agent is limited to 25 rather than 100 steps per episode. Importantly, VDO induces the MDL-C default policy to ignore input features which are, on average, less predictive of the control policy's behavior. In this case, the default policy learns to ignore the goal feature and the reward obtained on the previous timestep. This is because, when averaging across goal locations, the agent's current position ($s_h$) and its previous direction ($a_{h-1}$) are more informative of its next action—typically, heading towards the nearest door. In contrast, the un-regularized default policy of the RPO agent does not drop these features (Section I for a visualization and Section H for more details). By learning to ignore the goals present in phase 1 and encoding useful behavior regardless of goal location, MDL-C's develops more effective regularization in phase 2, enabling it to adapt more quickly than other methods (Fig. 5.1c, top), particularly RPO, which overfits to phase 1's goals. ManualIA also adapts quickly, as its default policy is hard-coded to ignore the goal feature.

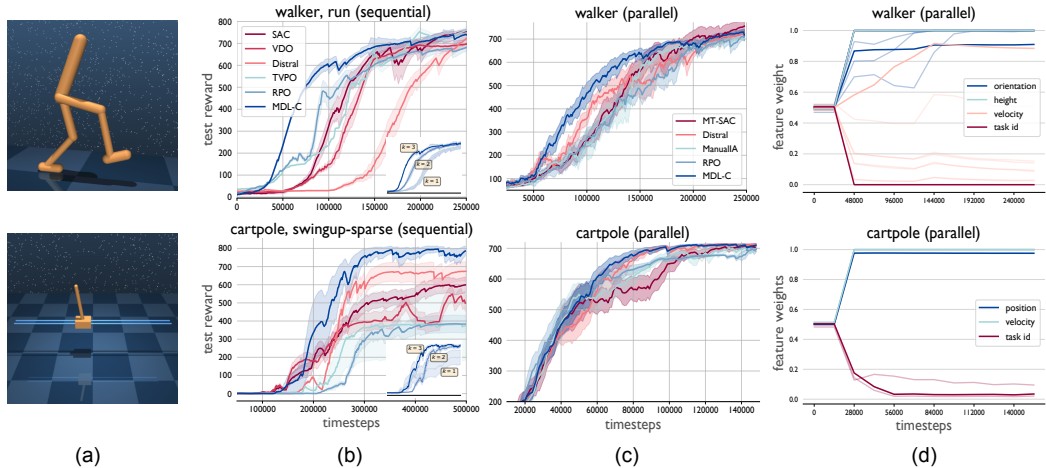

(a)  (b)  (c)  (d)

Figure 5.2: MDL-C improves both sequential and parallel learning in continuous control tasks. All curves represent averages taken over 8 random seeds, with the shading indicating standard error. In (b), insets show the improvement of MDL-C as $k$ increases, and in (d), solid curves represent averages over each feature within a category.

**Robustness to Rule Changes**   In this setting, there are only two possible goal locations, one at the top left of the environment, and the other at the bottom right. In training phase 1, the agent receives a goal feature as input which indicates the state index of the rewarded location for that episode. In phase 2, the goal feature switches from marking the reward location to marking the unrewarded location. That is, if the reward is in the top left, the goal feature will point to the bottom right. Here, the danger for the agent isn't overfitting to a particular goal or goals, but rather "overfitting" to the reward-based rules associated with a given feature. As we saw in Fig. 5.1c (top), an un-regularized default policy, will copy the control policy and overfit to a particular setting. Fortunately, the MDL-C default policy learns to ignore features which are, on average, less useful for predicting the control policy's behavior—the goal and previous reward features. This renders the agent more robust to contingency switches like the one described, as we can see in Fig. 5.1c (bottom). These examples illustrate that MDL-C enables agents to effectively learn the consistent structure of a group of tasks, regardless of its semantics, and "compress out" information which is less informative on average.

## 5.2   CONTINUOUS CONTROL

A more challenging application area is that of high-dimensional continuous control. In this setting, we presented agents with multitask learning problems using environments from the DeepMind Control Suite (DMC; (Tassa et al., 2018)). We used soft actor critic (SAC; (Haarnoja et al., 2018)) as the base agent. We tested MDL-C in both the sequential and parallel settings on two domains from DMC: walker and cartpole (Fig. 5.2a). Additional details can be found in Section H.

**Sequential Tasks**   In the sequential setting, tasks are sampled one at a time uniformly without replacement from the available tasks within each domain, with the default policy distribution conserved across tasks. For walker, these tasks are `stand`, `walk`, and `run`. In `stand`, the agent is rewarded for increasing the height of its center of mass, and in the latter two tasks, an additional reward is given for forward velocity. For cartpole, there are four tasks: `balance`, `balance-sparse`, `swingup`, and `swingup-sparse`. In the `balance` tasks, the agent must keep a rotating pole upright, and in the `swingup` tasks, it must additionally learn to swing the pole upwards from an initial downward orientation. Performance results for the hardest task within each domain (`run` in walker and `swingup-sparse` in cartpole) for each method are plotted in Fig. 5.2b, where $k$ indicates the task round at which the task was sampled. We can see that as $k$ increases (as more tasks have been seen previously), MDL-C's performance improves. Importantly, the RPO agent's default policy, which is un-regularized, overfits to the previous task, essentially copying the optimal policy's behavior. This can severely hinder the agent's performance when the subsequent task requires different behavior. For example, on `swingup-sparse`, if the previous task is `swingup`, the RPO agent performs well,

as the goal is identical. However, if the previous task is `balance` or `balance-sparse`, the agent never learns to swing the pole upwards, significantly reducing its average performance. Another important point to note is that because the agent is not given an explicit goal feature in this setting, methods like MANUALIA which rely on prior knowledge about the agent's inputs cannot be applied.

**Parallel Tasks**   We also tested parallel-task versions of SAC, MANUALIA, and MDL-C based on the model of Yu et al. (2019). In this framework, a task within each domain is randomly sampled at the start of each episodeand the agent learns a single control policy for all tasks. Performance is plotted in Fig. 5.2c, where we can again see that MDL-C accelerates convergence relative to the baseline methods. This marks a difference compared to the easier FourRooms environment, in which MDL-C and MANUALIA performed roughly the same. As before, one clue to the difference can be found in the input features that the MDL-C default policy chooses to ignore (Fig. 5.2d). For walker, inputs are 24-dimensional, with 14 features related to the joint orientations, 1 feature indicating the height of the agent's center of mass, and 9 features indicating velocity components. For cartpole, there are 5 input dimensions, with 3 pertaining to position and 2 to velocity. In the walker domain, where the performance difference is greatest, the MDL-C agent not only ignores the added task ID feature, but also the several features related to velocity. In contrast, in the cartpole domain, MDL-C only ignores the task ID feature, just as MANUALIA does, and the performance gap is smaller. This illustrates that MDL-C learns to compress out spurious information even in settings for which it is difficult to identify *a priori*. In order to test the effect of the learned asymmetry on performance more directly, we implemented a variant of MANUALIA in which all of the features which MDL-C learned to ignore were manually hidden from the default policy (Fig. I.4). Interestingly, while this method improved over standard MANUALIA, it didn't completely close the gap with MDL-C, indicating there are downstream effects within the network beyond input processing which are important for the default policy's effectiveness. We hope to explore these effects in more detail in future work.

## 6   RELATED WORK

MDL-C can be viewed as an extension of recent approaches to learning default policies ("behavioral priors") from the optimal policies of related tasks (Teh et al., 2017; Tirumala et al., 2020). For a default policy to be useful for transfer learning, it is crucial to balance the ability of the default policy to "copy" the control policies with its expressiveness. If the default policy is too expressive, it is likely to overfit on past tasks and fail to generalize to unseen tasks. Whereas prior work primarily hand-crafts structural constraints into the default policies to avoid overfitting (e.g., by hiding certain state information from the default policy; Galashov et al., 2019), MDL-C learns such a balance from data with sparsity-inducing priors via variational inference. MDL-C may also be derived from the RL-as-inference framework (Levine, 2018; Section A). MDL-C thus has close connections with algorithms such as MPO (Abdolmaleki et al., 2018) and VIREL (Fellows et al., 2020), discussed in Section A. As a general framework, MDL-C is also connected to the long and well-established literature on choosing appropriate Bayesian priors (Jeffreys, 1946; Bernardo, 2005; Casella, 1985), and more recent work that focuses on learning such priors for large-scale machine learning models (Nalisnick and Smyth, 2017; Nalisnick et al., 2021; Atanov et al., 2018). For a further discussion of related work, particularly concerning the application of MDL to the RL setting, see Section C.

## 7   CONCLUSION

Inspired by dual process theories and the MDL principle, we propose a regularized policy optimization framework for multitask RL which aims to learn a simple default policy encoding a low-complexity distillation of the optimal behavior for some family of tasks. By encouraging the default policy to maintain a low effective description length, MDL-C ensures that it does not overfit to spurious correlations among the (approximately) optimal policies learned by the agent. We described MDL-C's formal properties and demonstrated its empirical effectiveness in discrete and continuous control tasks. There are of course limitations of MDL-C, which we believe represent opportunities for future work (see Section E). Promising research directions include integrating MDL-C with multitask RL approaches which balance a larger set of policies (Barreto et al., 2020; Moskovitz et al., 2022b; Thakoor et al., 2022) as well considering nonstationary environments (Parker-Holder et al., 2022). We hope MDL-C inspires further advances in multitask RL.

**Acknowledgements**   The authors would like to thank Kevin Miller, DJ Strouse, Marcel Binz, and Alexander Galashov for useful discussions and suggested improvements to the manuscript. Work funded by the Gatsby Charitable Foundation.

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

# Minimum Description Length Control
## Supplementary Information

## A  REINFORCEMENT LEARNING AS INFERENCE

The control as inference framework (Levine, 2018) associates every time step $h$ with a binary "optimality" random variable $\mathcal{O}_h \in \{0, 1\}$ that indicates whether $a_h$ is optimal at state $s_h$ ($\mathcal{O}_h = 1$ for optimal, and $\mathcal{O}_h = 0$ for not). The optimality variable has the conditional distribution $P(\mathcal{O}_h = 1|s_h, a_h) = \exp(r(s_h, a_h))$, which scales exponentially with the reward received taking action $a_h$ in state $s_h$.

Denote $\mathcal{O}_H$ as the event that $\mathcal{O}_s = 1$ for $s = 0, \dots, H - 1$. Then the log-likelihood that a policy $\pi_w(a|s)$ is optimal over a horizon $H$ is given by:

$$P(\mathcal{O}_H) = \int P(\mathcal{O}_H|\tau) P^{\pi_w}(\tau|w) p(w) d\tau dw.$$

By performing variational inference, we can lower-bound the log-likelihood with the ELBO:

$$\log P(\mathcal{O}_H) \geq \mathbb{E}_{\nu_\pi(\tau)} \sum_{h=0}^{H-1} \left( r(s_h, a_h) - \mathbb{E}_{\nu_\theta(w)} \mathsf{KL}[\pi_\theta(a_h|s_h), \pi_w(a_h|s_h)] \right) \\ - \mathsf{KL}[\nu_\phi(w), p(w)],$$

(A.1)

where $\nu_{\theta,\phi}(\tau, w) = \nu_\theta(\tau)\nu_\phi(w)$ is the variational posterior,

$$\nu_\theta(\tau) = \rho(s_0) \prod_{h=0}^{H-1} P(s_{h+1}|s_h, a_h)\pi_\theta(a_h, s_h)$$

and $\{\theta, \phi\}$ are the variational parameters. We can maximize this objective iteratively by performing coordinate ascent on $\{\theta, \phi\}$:

$$\theta \leftarrow \theta + \eta \nabla_\theta \left( \mathbb{E}_{\nu_\theta(\tau)} \sum_{h=0}^{H-1} \left( r(s_h, a_h) - \mathbb{E}_{\nu_\theta(w)} \mathsf{KL}[\pi_\theta(a_h|s_h), \pi_w(a_h|s_h)] \right) \right),$$

(A.2)

$$\phi \leftarrow \phi - \eta \nabla_\phi \left( \mathbb{E}_{\nu_\theta(\tau)} \sum_{h=0}^{H-1} \mathbb{E}_{\nu_\theta(w)} \mathsf{KL}[\pi_\theta(a_h|s_h), \pi_w(a_h|s_h)] + \mathsf{KL}[\nu_\phi(w), p(w)] \right)$$

(A.3)

where $\eta$ is a learning rate parameter. Note that Equation (A.3) is equivalent to Equation (4.1) and Equation (G.8), and Equation (A.2) is equivalent to Equation (G.7) with the KL reversed.

**Connection to Maximum a Posteriori Policy Optimization (MPO)**   MDL-C is closely related to MPO (Abdolmaleki et al., 2018), with three key differences. First, MDL-C performs variational inference on the parameters of the default policy with an approximate posterior $\nu_\phi(w)$, whereas MPO performs MAP inference. Second, MPO places a normal prior on $w$, which in effect penalizes the L2 norm of $w$. In contrast, MDL-C uses sparsity-inducing priors such as the normal-Jeffreys prior. Third, MDL-C uses a parametric $\pi_\theta$, whereas MPO uses a non-parametric one[2]. While there is also a parametric variant of MPO, this variant does not maintain $\theta$ and $\phi$ separately. Instead, this variant directly sets $\theta$ to $\phi$ in Equation (A.2). This illustrates the key conceptual difference between MDL-C and MPO. MDL-C makes a clear distinction between the control policy $\pi_\theta$ and the default policy $\pi_w$, with the two policies serving two distinct purposes: the control policy for performing on the current task, the default policy for distilling optimal policies across tasks and generalizing to new ones. MPO, on the other hand, treats $\pi_\theta$ and $\pi_w$ as fundamentally the same object.

Like MPO, VIREL (Fellows et al., 2020) can be derived from the control as inference framework. In fact, Fellows et al. showed that a parametric variant of MPO can be derived from VIREL (Fellows et al., 2020). The key novelty that sets VIREL apart from both MPO and MDL-C is an adaptive temperature parameter that dynamically updates the influence of the KL term in Equation (A.2).

---

[2]In practice, MPO parametrizes $\pi_\theta$ implicitly with a parameterized action-value function and the default policy.

## B    MULTITASK RL FRAMEWORKS

We believe the objective which best captures naturalistic settings is the average reward obtained over the agent's "lifetime": $\lim_{T\to\infty} \frac{1}{T}\mathbb{E}\sum_{t=1}^{T} r(s_t, a_t)$. Typical objectives include finding either a single policy or a set of policies which maximize worst- or average-case value: $\max_\pi \min_{M\in\mathcal{M}} V_M^\pi$ (Zahavy et al., 2021) or $\max_\pi \mathbb{E}_{\mathsf{P}_\mathcal{M}} V_M^\pi$ (Moskovitz et al., 2022a). When the emphasis is on decreasing the required sample complexity of learning new tasks, a useful metric is *cumulative regret*: the agent's total shortfall across training compared to an optimal agent. In practice, it's often simplest to consider the task distribution $\mathsf{P}_\mathcal{M}$ to be a categorical distribution defined over a discrete set of tasks $\mathcal{M} \coloneqq \{M_k\}_{k=1}^K$, though continuous densities over MDPs are also possible. Two multitask settings which we consider here are *parallel task* RL and *sequential task* RL. In typical parallel task training (Yu et al., 2019), a new MDP is sampled from $\mathsf{P}_\mathcal{M}$ at the start of every episode and is associated with a particular input feature $g \in \mathcal{G}$ that indicates to the agent which task has been sampled. The agent's performance is evaluated on all tasks $M \in \mathcal{M}$ together. In the *sequential* task setting (Moskovitz et al., 2022a; Pacchiano et al., 2022), tasks (MDPs) are sampled one at a time from $\mathsf{P}_\mathcal{M}$, with the agent training on each until convergence. In contrast to continual learning (Kessler et al., 2021), the agent's goal is simply to learn a new policy for each task more quickly as more are sampled, rather than learning a single policy which maintains its performance across tasks. Another important setting is *meta-RL*, which we do not consider here. In the meta-RL setting, the agent trains on each sampled task for only a few episodes each with the goal of improving few-shot performance and is meta-tested on a set of held-out tasks (Yu et al., 2019; Finn et al., 2017).

Another strain of work in multitask RL assumes some form of shared structure in the transition dynamics (Pacchiano et al., 2022; ?; ?). Specifically, the core assumption made by these works is that the transition dynamics are linearly decodable from a set of features which is shared across tasks or in which the transition matrix admits a low-rank decomposition. This is very different from our own structural assumption—that is, in its simplest form, that the optimal policies of the tasks with which our agents are faced take similar actions in at least some part of the state space. Beyond this, the MDPs in $\mathcal{M}$ need only share the same state and action space, with no direct assumptions about transitions or rewards. This is important, because the assumed structures in the transition distribution made by Pacchiano et al. (2022); ?); ? act as the starting points for algorithm development. MDL-C/RPO/TVPO however, can leverage similarity among optimal policies when it exists, but are not dependent on it as a prerequisite. (E.g., TVPO (and RPO/MDL-C) is guaranteed to perform no worse than log-barrier regularization, which has a polynomial sample complexity guarantee.) Ideally, we'd like a generalist method which can identify on its own and exploit different types of structure in the environment.

## C    ADDITIONAL RELATED WORK

Previous work has also applied the MDL principle in an RL context, though primarily in the context of unsupervised skill learning (Zhang et al., 2021; Thrun and Schwartz, 1994). For example, Thrun and Schwartz (1994) are concerned with a set of "skills" which are policies defined only over a subset of the state space that are reused across tasks. They consider tabular methods, measuring a pseudo-description length as

$$DL = \sum_{s\in\mathcal{S}} \sum_{M\in\mathcal{M}} P_M^*(s) + \sum_{n\in N} |S_n|, \tag{C.1}$$

where $P_M^*(s)$ is the probability that no skill selects an action in state $s$ for task $M$ and the agent must compute the optimal $Q$-values in state $s$ for $M$, $N$ is the number of skills, and $|S_n|$ is the number states for which skill $n$ is defined. They then trade off this description length term with performance across a series of tabular environments.

One other related method is DISTRAL (Teh et al., 2017), which uses the following objective in the parallel task setting:

$$\mathcal{J}^{\text{Distral}}(\theta, \phi) = V^{\pi_\theta} - \mathbb{E}_{s\sim d^{\pi_\theta}}\left[\alpha\mathsf{KL}[\pi_\theta(\cdot|s), \pi_\phi(\cdot|s)] + \beta\mathsf{H}[\pi_\theta(\cdot|s)]\right]. \tag{C.2}$$

That is, like the un-regularized RPO method, DISTRAL can be seen as performing maximum-likelihood estimation to learn the (unconstrained) default policy, while adding an entropy bonus to the control policy.

Another important method in the sequential setting is TVPO (Moskovitz et al., 2022a), in which (in the tabular case) the default policy is defined as a softmax over the average action frequencies of the optimal policies for the tasks that the agent has seen so far. That is, if the average optimal action in a state $s$ is given by

$$\hat{\xi}_k(s, a) = \frac{1}{k} \sum_{i=1}^{k} \mathbb{1}(\pi_i^\star(s) = a),$$

then the TVPO default policy is

$$\pi_w(a|s) = \text{softmax}\left(\hat{\xi}_k(s, a)/\beta(k)\right),$$

where $\beta(k)$ is a temperature which decays as $k \to \infty$. In high-dimensional state and action spaces, this tabular solution can be approximated by training a default policy to predict the converged control policy's actions in each task. Importantly, this is equivalent to using KL distillation in that the default policies will converge to the same barycenter policy (Moskovitz et al., 2022a), as long as the distillation is only performed once the control policy has converged in each task. Using KL distillation in this way is exactly the RPO baseline that we use in this paper. Crucially, the use of the softmax with decaying temperature was introduced by Moskovitz et al. (2022a) as a useful 'hack' to prevent the default policy from overfitting to early tasks, as the optimal default policy is the barycenter policy (approximated as the number of draws from the task distribution grows). Thus, MDL-C can itself be seen as a scalable advancement of TVPO which models the agent's epistemic uncertainty about the task distribution by placing a sparse prior over the default policy parameters (and uses a distillation loss rather than action prediction). In other words, MDL-C represents a principled approach to reducing the risk of default policy overfitting in the low-data regime.

Finally, Brunskill and Li (2013) consider a similar training and task structure to our own, but use a model-based approach to learn the underlying MDPs.

## D  MOTIVATING THE CHOICE OF SPARSITY-INDUCING PRIORS

As a reminder, the generative model of optimal parameters in Section 4.1 is given by:

$$\overline{w}_i|\beta, \sigma^2 \sim \mathcal{N}(0, \frac{1-\beta}{\beta}\sigma^2 I_d), \tag{D.1}$$

$$w_{ik}|\overline{w}_i, \sigma^2, \beta \sim \mathcal{N}(\overline{w}, \sigma^2 I_d) \tag{D.2}$$

with marginal and posterior densities

$$p(w_{ik}|\sigma^2, \beta) = \mathcal{N}(0, \sigma^2 \beta^{-1} I_d), \tag{D.3}$$

$$p(\overline{w}_i|w_{ik}, \sigma^2, \beta) = \mathcal{N}\left((1-\beta)w_{ik}, (1-\beta)\sigma^2 I_d\right). \tag{D.4}$$

In the rest of this section, we set $\sigma^2 = 1$ for simplicity and drop the indices on $w$ and $\overline{w}$ to remove clutter.

### D.1  CORRESPONDENCE BETWEEN $p(z)$ AND $p(\beta)$

In Section 4.1, we draw a connection between $p(\beta) \propto \beta^{-1}$ and the normal-Jeffreys prior, which is commonly used for compressing deep neural networks (Louizos et al., 2017). In Table 1, we expand on this connection and list $p(\beta)$ for two other commonly-used priors for scale mixture of normal distributions: Jeffreys, Inverse-gamma, and Inverse-beta. Note that the half-Cauchy distribution $p(z) \propto (1 + z^2)^{-1}$ is a special case of the inverse-beta distribution for $s = t = 1/2$. Half-cauchy prior is another commonly used prior for compressing Bayesian neural networks (Louizos et al., 2017).

### D.2  MSE RISK

In this section, we prove that the Bayes estimators for the Jeffreys, inverse-gamma, and the inverse-beta (by extension the half-Cauchy) distributions dominate the maximum-likelihood estimator with respect to the mean-squared error.

Define the mean-squared error of an estimator $\hat{w}(x)$ of $\overline{w}$ as

$$\text{MSE}(\overline{w}, \hat{w}) = \mathbb{E}_x \|\hat{w}(x) - \overline{w}\|^2, \tag{D.5}$$

where the expectation is taken over $\mathcal{N}(x; \overline{w}, \alpha^2)$. Immediately, we have $\text{R}(\overline{w}, \hat{w}^{(\text{ML})}) = d$, where $\hat{w}^{(\text{ML})}(x) = x$ is the maximum-likelihood estimator. An estimator $\hat{w}^{(a)}(x)$ is said to dominate another estimator $\hat{w}^{(b)}(x)$ if $\text{MSE}(\overline{w}, \hat{w}_a) \leq \text{MSE}(\overline{w}, \hat{w}_b)$ for all $\overline{w}$ and the inequality is strict for a set of positive Lesbesgue measure. It is well-known that the maximum-likelihood estimator is minimax (George et al., 2006), and thus any estimator that dominates the maximum-likelihood estimator is also minimax.

To compute the mean-squared error risk for an estimator $\hat{w}(x)$, observe that

$$\|\hat{w}(x) - \overline{w}\|^2 = \|x - \hat{w}(x)\|^2 - \|x - \overline{w}\|^2 + 2(\hat{w}(x) - \overline{w})^\top (x - \overline{w}). \tag{D.6}$$

Taking expectations on both sides gives

$$\text{MSE}(\overline{w}, \hat{w}) = \mathbb{E}_x \|x - \hat{w}(x)\|^2 - d + 2 \sum_{i=1}^d \text{Cov}(\hat{w}_i(x), x_i) \tag{D.7}$$

$$= \mathbb{E}_x \|x - \hat{w}(x)\|^2 - d + 2\mathbb{E}_x \nabla \cdot \hat{w}(x) \tag{D.8}$$

where $\nabla = (\partial/\partial x_1, \ldots, \partial/\partial x_d)$ and we apply Stein's lemma $\text{cov}(\hat{w}_i(x), x_i) = \mathbb{E}_x \partial \hat{w}_i / \partial x_i$ in the last line. If the estimator takes the form $\hat{w}(x) = x + \gamma(x)$, the expression simplifies as:

$$\text{MSE}(\overline{w}, \hat{w}) = d + \mathbb{E}_x \|\gamma(x)\|^2 + 2\mathbb{E}_x \nabla \cdot \gamma(x). \tag{D.9}$$

Therefore, an estimator $\hat{w}(x) = x + \gamma(x)$ dominates $\hat{w}^{(\text{ML})}(x)$ if

$$\text{MSE}(\overline{w}, \hat{w}) - \text{MSE}(\overline{w}, \hat{w}^{(\text{ML})}) = \mathbb{E}_x \left[ \|\gamma(x)\|^2 + 2\nabla \cdot \gamma(x) \right] \leq 0 \tag{D.10}$$

for all $\overline{w}$ and the inequality is strict on a set of positive Lesbesgue measure.

### D.2.1 JAMES-STEIN ESTIMATOR

The famous Jame-Stein estimator is defined as

$$\hat{w}^{(\text{JS})}(x) = x + \gamma^{(\text{JS})}(x), \quad \gamma^{(\text{JS})}(x) = -(d-2)x/\|x\|^2, \tag{D.11}$$

with

$$\nabla \cdot \gamma^{(\text{JS})}(x) = \sum_{i=1}^d \left[ -\frac{d-2}{\|x\|^2} + 2\frac{d-2}{(\|x\|^2)^2} x_i^2 \right] = -\frac{(d-2)^2}{\|x\|^2}, \tag{D.12}$$

$$\|\gamma^{(\text{JS})}(x)\|^2 = \frac{(d-2)^2}{\|x\|^2}. \tag{D.13}$$

Substituting $\nabla \cdot \gamma^{(\text{JS})}(x)$ and $\|\gamma^{(\text{JS})}(x)\|^2$ into Equation (D.10), we have

$$\text{MSE}(\overline{w}, \hat{w}^{(\text{JS})}) - \text{MSE}(\overline{w}, \hat{w}^{(\text{ML})}) = \mathbb{E}_x \frac{(d-2)^2}{\|x\|^2}. \tag{D.14}$$

Thus, the James-Stein estimator dominates the maximum-likelihood estimator for $d > 2$.

| Prior name | $p(z^2)$ | $p(\beta)$ |
|---|---|---|
| Jeffreys | $p(z^2) \propto z^{-2}$ | $p(\beta) \propto \beta^{-1}$ |
| Inverse-gamma | $p(z^2) \propto z^{-2(s+1)} e^{-t/(2z^2)}$ | $p(\beta) \propto \beta^{s-1} e^{-t\beta/2}$ |
| Inverse-beta | $p(z^2) \propto (z^2)^{t-1}(1+z^2)^{-(s+t)}$ | $p(\beta) \propto \beta^{-(s+2t+1)}(1+\beta)^{-(s+t)}$ |

Table 1: Correspondence between $p(z^2)$ and $p(\beta)$.

### D.2.2 BAYES ESTIMATORS

The Bayes estimator for a prior choice $p(\beta)$ is given by (**?**):

$$\hat{w}^{(p(\beta))}(x) = x + \gamma^{(p(\beta))}(x), \quad \gamma^{(p(\beta))}(x) = \nabla \log m(x), \tag{D.15}$$

where

$$m(x) = \int \mathcal{N}(x; 0, \beta^{-1}I_d)p(\beta)d\beta \tag{D.16}$$

$$= \int (2\pi)^{-\frac{1}{2}}\beta^{d/2}\exp\left(-\beta x^2/2\right)p(\beta)d\beta. \tag{D.17}$$

Substituting $\gamma^{(p(\beta))}(x)$ into Equation (D.10), we find that the condition for the Bayes estimator to be minimax is given by (George et al., 2006):

$$\text{MSE}(\overline{w}, \hat{w}^{(\text{B})}) - \text{MSE}(\overline{w}, \hat{w}^{(\text{ML})}) = \mathbb{E}_x\left[-\|\nabla \log m(x)\|^2 + 2\frac{\nabla^2 m(x)}{m(x)}\right] \tag{D.18}$$

$$= \mathbb{E}_x\left[4\frac{\nabla^2 \sqrt{m(x)}}{\sqrt{m(x)}}\right] \leq 0, \tag{D.19}$$

where $\nabla^2 = \sum_i \partial^2/\partial x_i^2$ is the Laplace operator. This condition holds when $\sqrt{m(x)}$ is superharmonic (i.e., $\sqrt{m(x)} \leq 0, \forall x \in \mathbb{R}^d$), suggesting a recipe for constructing Bayes estimators that dominate the maximum likelihood estimator, summarized in the following proposition.

**Proposition D.1** (Extension of Theorem 1 in Fourdrinier et al., 1998). *Let $p(\beta)$ be a positive function such that $f(\beta) = \beta p'(\beta)/p(\beta)$ can be decomposed as $f_1(\beta) + f_2(\beta)$ where $f_1$ is non-decreasing, $f_1 \leq A$, $0 < f_2 \leq B$, and $A/2 + B \leq (d-6)/4$. Assume also that $\lim_{\beta \to 0} \beta^{d/2+2}p(\beta) = 0$. Then, $\nabla^2\sqrt{m(x)} \leq 0$ and the Bayes estimator is minimax. If $A/2 + B < (d-6)/4$, then the Bayes estimator dominates $\hat{w}^{(ML)}(x)$.*

*Proof.* This proof largely follows the proof of Theorem 1 in (Fourdrinier et al., 1998).

Note that Equation (D.18) holds if

$$\nabla^2\sqrt{m(x)} = \frac{1}{2\sqrt{m(x)}}\left(\nabla^2 m(x) - \frac{1}{2}\frac{\|\nabla m(x)\|^2}{m(x)}\right) \leq 0 \quad \forall x \in \mathbb{R}^d, \tag{D.20}$$

or equivalently

$$\frac{\nabla^2 m(x)}{\|\nabla m(x)\|} - \frac{1}{2}\frac{\|\nabla m(x)\|}{m(x)} \leq 0 \quad \forall x \in \mathbb{R}^d. \tag{D.21}$$

Computing the derivatives, we get the condition

$$\frac{\int_0^1 \left(\beta\|x\|^2 - d\right)\beta^{d/2+1}e^{-\beta\|x\|^2/2}p(\beta)d\beta}{\|x\|\int_0^1 \beta^{d/2+1}e^{-\beta\|x\|^2/2}p(\beta)d\beta} - \frac{1}{2}\frac{\|x\|\int_0^1 \beta^{d/2+1}e^{-\beta\|x\|^2/2}p(\beta)d\beta}{\int_0^1 \beta^{d/2}e^{-\beta\|x\|^2/2}p(\beta)d\beta} \leq 0. \tag{D.22}$$

Divide both sides by $\|x\|$ and rearrange to get

$$\frac{\int_0^1 \beta^{d/2+2}e^{-\beta\|x\|^2/2}p(\beta)d\beta}{\int_0^1 \beta^{d/2+1}e^{-\beta\|x\|^2/2}p(\beta)d\beta} - \frac{1}{2}\frac{\int_0^1 \beta^{d/2+1}e^{-\beta\|x\|^2/2}p(\beta)d\beta}{\int \beta^{d/2}e^{-\beta\|x\|^2/2}p(\beta)d\beta} \leq \frac{d}{\|x\|^2}. \tag{D.23}$$

Next, we integrate by parts the numerator of the first term on the left-hand side to get:

$$\int_0^1 \beta^{d/2+2}e^{-\beta\|x\|^2/2}p(\beta)d\beta = -\frac{2}{\|x\|^2}\left[\beta^{d/2+2}e^{-\beta\|x\|^2/2}p(\beta)\right]_0^1 \tag{D.24}$$

$$+ \frac{d+4}{\|x\|^2}\int_0^1 \beta^{d/2+1}e^{-\beta\|x\|^2/2}p(\beta)d\beta$$

$$+ \frac{2}{\|x\|^2}\int_0^1 \beta^{d/2+2}e^{-\beta\|x\|^2/2}p'(\beta)d\beta,$$

where the middle term is the same as the denominator of the first term in Equation (D.23). Integrating by parts the second term gives the same expression as that of the first term, but with $d - 2$ in place of $d$ everywhere. Substituting these expressions back into Equation (D.23), collecting like terms, and dividing both sides by $2/\|x\|^2$, gives:

$$
\frac{\int_0^1 \beta^{d/2+2} e^{-\beta\|x\|^2/2} p'(\beta) d\beta}{\int_0^1 \beta^{d/2+1} e^{-\beta\|x\|^2/2} p(\beta) d\beta} - \frac{1}{2} \frac{\int_0^1 \beta^{d/2+1} e^{-\beta\|x\|^2/2} p'(\beta) d\beta}{\int_0^1 \beta^{d/2} e^{-\beta\|x\|^2/2} p(\beta) d\beta} + \kappa_0 + \kappa_1 \tag{D.25}
$$

$$
\leq \frac{d}{2} - \frac{d+4}{2} + \frac{1}{2}\frac{d+2}{2} = \frac{d-6}{4},
$$

where

$$
\kappa_1 = -\frac{\lim_{\beta \to 1} \beta^{d/2+2} e^{-\beta\|x\|^2/2} p(\beta)}{\int_0^1 \beta^{d/2+1} e^{-\beta\|x\|^2/2} p(\beta) d\beta} + \frac{1}{2}\frac{\lim_{\beta \to 1} \beta^{d/2+1} e^{-\beta\|x\|^2/2} p(\beta)}{\int_0^1 \beta^{d/2} e^{-\beta\|x\|^2/2} p(\beta) d\beta}, \tag{D.26}
$$

$$
\kappa_0 = \frac{\lim_{\beta \to 0} \beta^{d/2+2} e^{-\beta\|x\|^2/2} p(\beta)}{\int_0^1 \beta^{d/2+1} e^{-\beta\|x\|^2/2} p(\beta) d\beta} - \frac{1}{2}\frac{\lim_{\beta \to 0} \beta^{d/2+1} e^{-\beta\|x\|^2/2} p(\beta)}{\int_0^1 \beta^{d/2} e^{-\beta\|x\|^2/2} p(\beta) d\beta}. \tag{D.27}
$$

Here, both $\kappa_0$ and $\kappa_1$ are nonpositive: (i) $\kappa_0$ is nonpositive because the first term vanishes due to the boundary conditions and the second term is nonpositive, and (ii) $\kappa_1$ is nonpositive because the limits of the numerators of the two terms are equal while the denominator of the second term is larger than that of the first. We can thus drop $\kappa_0$ and $\kappa_1$ to get the sufficient condition:

$$
\mathbb{E}_d(f) - \frac{1}{2}\mathbb{E}_{d-2}(f) \leq \frac{d-6}{4}, \tag{D.28}
$$

where $\mathbb{E}_d$ denotes expectation with respect to the density

$$
g_d(\beta) = \frac{\beta^{d/2+1} e^{-\beta\|x\|^2/2} p(\beta)}{\int_0^1 \beta^{d/2+1} e^{-\beta\|x\|^2/2} p(\beta) d\beta} \tag{D.29}
$$

and where $f(\beta) = \beta p'(\beta)/p(\beta)$.

Because $g_d(\beta)$ is a family of monotone increasing likelihood ratio in $d$ and $f_1$ is nonincreasing and bounded by $A$, we have $\mathbb{E}_d(f_1) - \mathbb{E}_{d-2}(f_1)/2 \leq A/2$. We have $\mathbb{E}_d(f_2) - \mathbb{E}_{d-2}(f_2)/2 \leq B$ because $0 < f_2 \leq B$. Taken together, we have

$$
\mathbb{E}_d(f) - \mathbb{E}_{d-2}(f)/2 \leq A/2 + B \leq (k-6)/4. \tag{D.30}
$$

When the inequality is strict (i.e., $A/2 + B < (k-6)/4$), then $\nabla^2 \sqrt{m(x)} < 0$ and the Bayes estimator dominates the maximum-likelihood estimator. $\qquad\square$

Checking whether a given $p(\beta)$ satisfy the conditions in Proposition D.1 may be tedious. The following corollary is useful for construction $p(\beta)$ that satisfies the conditions in Proposition D.1.

**Corollary D.1** (Extension of Corollary 1 in Fourdrinier et al., 1998)**.** *Let $\psi$ be a continuous function that can be decomposed as $\psi_1 + \psi_2$, with $\psi_1 \leq C$, $\psi_1$ non-decreasing, $0 < \psi_2 \leq D$, and $C/2 + D \leq 0$. Let*

$$
p(\beta) = \exp\left(\frac{1}{2}\int_{\beta_0}^\beta \frac{2\psi(u) + d - 6}{u} du\right) \quad \forall \beta_0 \geq 0, \tag{D.31}
$$

*such that $\lim_{\beta \to 0} \beta^{d/2+2} p(\beta) = 0$ and $\beta_0 \in (0, 1)$ is a constant. Then, $p(\beta)$ results in a minimax Bayes estimator, which dominates the maximum likelihood estimator when $C/2 + D < 0$.*

*Proof.* The proof is the same as that of Corollary 1 in Fourdrinier et al., 1998, with Proposition D.1 in place of Theorem 1 in Fourdrinier et al., 1998. $\qquad\square$

Using Corollary D.1, we now check that the three priors listed in Table 1 and referenced in Section 4.1 lead to Bayes estimators that dominate the maximum-likelihood estimator.

**Jeffreys prior**   Let $\psi_1(u) = a$ for $a \leq 0$ and $\psi_2(u) = 0$. We have

$$p(\beta) = \exp\left(\frac{1}{2}\int_{\beta_0}^{\beta} \frac{2a + d - 6}{u} du\right) \propto \beta^{a + (d-6)/2}. \tag{D.32}$$

To satisfy $\lim_{\beta \to 0} \beta^{d/2+2} p(\beta) = 0$, we require $1 - d < a \leq 0$. We recover the improper normal-Jeffreys prior $p(\beta) \propto \beta^{-1}$, for $a = 2 - d/2$. The corresponding Bayes estimator dominates the maximum likelihood estimator when $d > 4$.

**Inverse-gamma prior**   Let $\psi_1(u) = a$ and $\psi_2(u) = b(1 - u)/2$ for $a \leq 0$ and $b \geq 0$. We have

$$p(\beta) = \exp\left(\int_{\beta_0}^{\beta} \frac{a + b(1 - u)/2 + (d - 6)/2}{u} du\right) \propto \beta^{a + (b+d-6)/2} e^{-b\beta/2}. \tag{D.33}$$

Setting $C = a$ and $D = b/2$, we get the followings conditions: $a + b \leq 0$ and $1 - d \leq a + b/2$. Note that when these conditions are met with $s = a + (b + d - 4)/2$ and $t = b$, we recover the inverse-gamma prior in Table 1.

**Inverse-beta (half-Cauchy) prior**   Let $\psi_1(u) = a$ and $\psi_2(u) = b/(u + 1)$ for $a \leq 0$ and $b \geq 0$. We have

$$p(\beta) = \exp\left(\int_{\beta_0}^{\beta} \frac{a + b/(1 + u) + (d - 6)/2}{u} du\right) \propto \beta^{a + b + (d-6)/2}(1 + \beta)^{-b}. \tag{D.34}$$

Setting $C = a$ and $D = b$, we get the condition $a/2 + b \leq 0$. To satisfy $\lim_{\beta \to 0} \beta^{d/2+2} p(\beta) = 0$, we require $1 - d < a + b \leq 0$. Note that this corresponds to the inverse-beta prior in Table 1 with $t = a + (d - 8)/2$ and $s = b - t$.

To recover the half-Cauchy prior, we set $b = 1$ and $a = (5 - d)/2$. All conditions in Corollary D.1 are satisfied when $d > 9$.

# E   LIMITATIONS

One weakness of the current theoretical analysis regarding the choice of sparsity-inducing priors is the assumption of Gaussian (and in particular, isotropic Gaussian) structure in the parameter space of optimal policies for clusters of tasks. In reality, there is likely a nontrivial degree of covariance among task parameterizations. Extending our analysis to more realistic forms of task structure is an important direction for future work. In a similar vein, the assumption that tasks are drawn iid from a fixed distribution is also unrealistic in naturalistic settings. It would be interesting to introduce some form of sequential structure (e.g., tasks are drawn from a Markov process). Another direction for future work is expanding beyond the "one control policy, one default policy" setup–having, for example, one default policy per task cluster and the ability to reuse and select (for example, using successor feature-like representations (Barreto et al., 2020; Barth-Maron et al., 2018; Moskovitz et al., 2022b)) among an actively-maintained set of control policies across tasks and task clusters would be useful.

# F   OCO BACKGROUND

In *online convex optimization* (OCO), the learner observes a series of convex loss functions $\ell_k : \mathsf{N} \to \mathbb{R}$, $k = 1, \ldots, K$, where $\mathsf{N} \subseteq \mathbb{R}^d$ is a convex set. After each round, the learner produces an output $x_k \in \mathsf{N}$ for which it will then incur a loss $\ell_k(x_k)$ (Orabona, 2019). At round $k$, the learner is usually assumed to have knowledge of $\ell_1, \ldots, \ell_{k-1}$, but no other assumptions are made about the sequence of loss functions. The learner's goal is to minimize its average regret:

$$\bar{\mathcal{R}}_K := \frac{1}{K}\sum_{k=1}^{K} \ell_k(x_k) - \min_{x \in \mathsf{N}} \frac{1}{K}\sum_{k=1}^{K} \ell_k(x). \tag{F.1}$$

One OCO algorithm which enjoys sublinear regret is *follow the regularized leader* (FTRL). In each round of FTRL, the learner selects the solution $x \in \mathsf{N}$ according to the following objective:

$$x_{k+1} = \underset{x \in \mathsf{N}}{\operatorname{argmin}} \, \psi_k(x) + \sum_{i=1}^{k-1} \ell_i(x), \tag{F.2}$$

where $\psi_k : \mathsf{N} \to \mathbb{R}$ is a convex regularization function.

# G  PROOFS OF PERFORMANCE BOUNDS AND ADDITIONAL THEORETICAL RESULTS

The following result is useful.

**Lemma G.1.** *The function $\ell(\nu) = \mathbb{E}_{w \sim \nu} f(w)$ is L-Lipschitz as long as $f : \mathcal{W} \to \mathbb{R}$ lies within $[0, L] \, \forall w \in \mathcal{W}$, where $\mathcal{W} \subseteq \mathbb{R}^d$ is a Hilbert space and $L < \infty$.*

*Proof.* We have

$$
\begin{aligned}
|\ell(\nu_1) - \ell(\nu_2)| &= |\mathbb{E}_{w \sim \nu_1} f(w) - \mathbb{E}_{w \sim \nu_2} f(w)| \\
&= \left| \int_{\mathcal{W}} (\nu_1(w) - \nu_2(w)) f(w) \, dw \right| \\
&= |\langle f, \nu_1 - \nu_2 \rangle_{\mathcal{W}}| \\
&\leq \|f\|_{\mathcal{W}} \|\nu_1 - \nu_2\|_{\mathcal{W}} \\
&\leq L \|\nu_1 - \nu_2\|_{\mathcal{W}},
\end{aligned}
$$

where the first inequality is due to Cauchy-Schwarz and the second is by assumption on $f$. $\qquad \square$

**Proposition G.1** (Default Policy Distribution Regret). *Let tasks $M_k$ be independently drawn from $\mathsf{P}_{\mathcal{M}}$ at every round, and let them each be associated with a deterministic optimal policy $\pi_k^\star : \mathcal{S} \to \mathcal{A}$. We make the following mild assumptions: i) $\pi_w(a^\star|s) \geq \epsilon > 0 \, \forall s \in \mathcal{S}$, where $a^\star = \pi_k^\star(s)$ and $\epsilon$ is a constant. ii) $\min_\nu \mathsf{KL}[\nu(\cdot), p(\cdot)] \to 0$ as $\mathrm{Var}[\nu] \to \infty$ for an appropriate choice of sparsity-inducing prior $p$. Then Algorithm 2 guarantees*

$$\mathbb{E}_{\mathsf{P}_{\mathcal{M}}}[\ell_K(\nu_K) - \ell_K(\bar{\nu}_K)] \leq (\mathbb{E}_{\mathsf{P}_{\mathcal{M}}} \mathsf{KL}[\bar{\nu}_K, p] + 1) \frac{\log(1/\epsilon)}{\sqrt{K}}. \tag{G.1}$$

*where $\bar{\nu}_K = \operatorname{argmin}_{\nu \in \mathsf{N}} \sum_{k=1}^{K} \ell_k(\nu)$.*

*Proof.* The first part of the proof sets up an application of Orabona (2019), Corollary 7.9.

To establish grounds for its application, we first note the standard result that the regularization functional $\psi(\nu) = \mathsf{KL}[\nu(w), p(w)]$ for probability measures $\nu, p \in \mathcal{P}(\mathcal{W})$ is 1-strongly convex in $\nu$ (Melbourne, 2020).

Finally, assumption (i) implies that the KL between the default policy and the optimal policy is upper-bounded: $\mathsf{KL}[\pi_k^\star, \pi_w] \leq \log 1/\epsilon$. Then by Lemma G.1, $\ell_k(\nu)$ is $L$-Lipschitz wrt the TV distance, where $L = \log 1/\epsilon$.

Note also that under a Gaussian parameterization for $\nu$, the distribution space $\mathsf{N}$ is the Gaussian parameter space $\mathsf{N} = \{(\mu, \Sigma) : \mu \in \mathbb{R}^d, \Sigma \in \mathbb{R}^{d \times d}, \Sigma \succeq 0\}$, which is convex (Boyd and Vandenberghe, 2004).

Then Orabona (2019), Corollary 7.9 gives

$$\frac{1}{K} \sum_{k=1}^{K} \ell_k(\nu_K) - \frac{1}{K} \sum_{k=1}^{K} \ell_k(\bar{\nu}_K) \leq \left( \frac{1}{\alpha} \mathsf{KL}[\bar{\nu}_K, p] + \alpha \right) \frac{L}{\sqrt{K}}, \tag{G.2}$$

where $\bar{\nu}_K = \operatorname{argmin}_\nu \sum_{k=1}^{K} \ell_k(\nu)$. The constant $\alpha \in \mathbb{R}^+$ is a hyperparameter, so we are free to set it to 1 (Orabona, 2019). Finally, we observe that $\mathbb{E}_{\mathsf{P}_{\mathcal{M}_i}} \frac{1}{K} \sum_{k=1}^{K} \ell(\nu_k) = \mathbb{E}_{\mathsf{P}_{\mathcal{M}_i}} \ell_K(\nu_K)$ and take the

expectation with respect to $\mathsf{P}_{\mathcal{M}_i}$ of both sides of Eq. (G.2) to get the desired result:

$$\mathbb{E}_{\mathsf{P}_{\mathcal{M}_i}}[\ell_K(\nu_K) - \ell_K(\bar{\nu}_K)] \leq (\mathbb{E}_{\mathsf{P}_{\mathcal{M}_i}}\mathsf{KL}[\bar{\nu}_K, p] + 1)\frac{L}{\sqrt{K}}. \tag{G.3}$$

$\square$

**Proposition 4.2** (Control Policy Sample Complexity). *Under the setting described in Proposition G.1, denote by $T_k$ the number of iterations to reach $\epsilon$-error for $M_k$ in the sense that $\min_{t \leq T_k}\{V^{\pi_k^*} - V^{(t)}\} \leq \epsilon$. whenever $t > T_k$. Further, denote the upper-bound in Eq. (G.1) by $G(K)$. In a finite MDP, from any initial $\theta^{(0)}$, and following gradient ascent, $\mathbb{E}_{M_k \sim \mathcal{P}_\mathcal{M}}[T_k]$ satisfies:*

$$\mathbb{E}_{M_k \sim \mathcal{P}_{\mathcal{M}_i}}[T_k] \geq \frac{80|\mathcal{A}|^2|\mathcal{S}|^2}{\epsilon^2(1-\gamma)^6}\mathbb{E}_{\substack{M_k \sim \mathcal{P}_{\mathcal{M}_i} \\ s \sim \mathrm{Unif}_\mathcal{S}}}\left[\kappa_{\mathcal{A}}^{\alpha_k}(s)\left\|\frac{d_\rho^{\pi_k^*}}{\mu}\right\|_\infty^2\right],$$

*where $\alpha_k(s) := d_{\mathrm{TV}}(\pi_k^\star(\cdot|s), \hat{\pi}_0(\cdot|s)) \leq \sqrt{G(K)}$, $\kappa_{\mathcal{A}}^{\alpha_k}(s) = \frac{2|\mathcal{A}|(1-\alpha(s))}{2|\mathcal{A}|(1-\alpha(s))-1}$, and $\mu$ is a measure over $\mathcal{S}$ such that $\mu(s) > 0 \ \forall s \in \mathcal{S}$.*

Note: In the above, there is a small error—it should be $\alpha_k(s) := \mathbb{E}_{w \sim \nu}d_{\mathrm{TV}}(\pi_k^\star(\cdot|s), \pi_w(\cdot|s)) \leq \sqrt{\frac{1}{2}G(K)}$. $d_\rho^\pi$ refers to the discounted state-occupancy distribution under $\pi$ with initial state distribution $\rho$:

$$d_\rho^\pi(s) = \mathbb{E}_{s_0 \sim \rho}(1-\gamma)\sum_{h \geq 0}\gamma^h\mathsf{P}^\pi(s_h = s|s_0). \tag{G.4}$$

Division between probability mass functions is assumed to be element-wise.

*Proof.* Without loss of generality, we prove the bound for a fixed state $s \in \mathcal{S}$, noting that the bound applies independently of our choice of $s$. We use the shorthand $\mathsf{KL}[\pi(\cdot|s), \pi_w(\cdot|s)] \to \mathsf{KL}[\pi, \pi_w]$ for brevity. We start by multiplying both sides of the bound from Proposition G.1 by $1/2$ and rearranging:

$$\frac{1}{2}\left(\mathbb{E}_{\mathsf{P}_{\mathcal{M}_i}}\ell_K(\bar{\nu}_K) + \frac{L}{\sqrt{K}}(\mathbb{E}_{\mathsf{P}_{\mathcal{M}_i}}\mathsf{KL}[\bar{\nu}_K, p] + 1)\right)$$

$$\geq \mathbb{E}_{\mathsf{P}_{\mathcal{M}_i}}\frac{1}{2}\ell_K(\nu_K)$$

$$= \mathbb{E}_{\mathsf{P}_{\mathcal{M}_i}}\mathbb{E}_{\nu_K}\frac{1}{2}\mathsf{KL}[\pi_K^\star, \pi_w]$$

$$\overset{(i)}{=} \mathbb{E}_{\mathsf{P}_{\mathcal{M}_i}}\left[\mathrm{Var}_{\nu_K}\left[\sqrt{\frac{1}{2}\mathsf{KL}[\pi_K^\star, \pi_w]}\right] + \mathbb{E}_{\nu_K}\left[\sqrt{\frac{1}{2}\mathsf{KL}[\pi_K^\star, \pi_w]}\right]^2\right] \tag{G.5}$$

$$\overset{(ii)}{\geq} \mathbb{E}_{\mathsf{P}_{\mathcal{M}_i}}\left[\mathbb{E}_{\nu_K}\left[\sqrt{\frac{1}{2}\mathsf{KL}[\pi_K^\star, \pi_w]}\right]^2\right]$$

where $(i)$ follows from the definition of the variance, and $(ii)$ follows from its non-negativity. We can rearrange to get

$$\frac{L}{2\sqrt{K}}(\mathbb{E}_{\mathsf{P}_{\mathcal{M}_i}}\mathsf{KL}[\bar{\nu}_K, p] + 1) \geq \mathbb{E}_{\mathsf{P}_{\mathcal{M}_i}}\mathbb{E}_{\nu_K}\left[\sqrt{\frac{1}{2}\mathsf{KL}[\pi_K^\star, \pi_w]}\right]^2 \tag{G.6}$$

$$\overset{(ii)}{\geq} \mathbb{E}_{\mathsf{P}_{\mathcal{M}_i}}\mathbb{E}_{\nu_K}[d_{\mathrm{TV}}(\pi_K^\star, \pi_w)]^2$$

where $(ii)$ follows from Pinsker's inequality. Letting $\alpha_K(s) = \sqrt{\frac{1}{2}G(K)}$ and applying Moskovitz et al. (2022a), Lemma 5.2 gives the desired result. $\square$

This upper-bound is signficant, as it shows that, all else being equal, a high complexity barycenter default policy distribution $\bar{\nu}_K$ (where complexity is measured by $\mathsf{KL}[\bar{\nu}_K, p]$) leads to a slower convergence rate in the control policy.

---

Algorithm 2: Idealized MDL-C for Multitask Learning

---

1: **require:** task distribution $\mathsf{P}_{\mathcal{M}}$, policy class $\Pi$, coefficients $\{\eta_k\}$
2: **initialize:** default policy distribution $\nu_1 \in \mathsf{N}$
3: **for** tasks $k = 1, 2, \ldots, K$ **do**
4:     Sample a task $M_k \sim \mathcal{P}_{\mathcal{M}}(\cdot)$
5:     Optimize control policy:

$$\hat{\pi}_k^\star = \underset{\pi \in \Pi}{\operatorname{argmax}} \, V_{M_k}^\pi - \lambda \mathbb{E}_{s \sim d^\pi} \mathbb{E}_{w \sim \nu_k} \mathsf{KL}[\pi_w(a|s), \pi(a|s)] \tag{G.7}$$

6:     Update default policy distribution:

$$\nu_{k+1} = \underset{\nu \in \mathsf{N}}{\operatorname{argmin}} \, \mathsf{KL}[\nu, p] + \mathbb{E}_{w \sim \nu} \mathsf{KL}[\hat{\pi}_k^\star, \pi_w] \tag{G.8}$$

7: **end for**

---

### G.1 MDL-C with Persistent Replay

Rather than rely on iid task draws to yield a bound on the expected regret under the task distribution, a more general formulation of MDL-C for sequential task learning is described in Algorithm 1. In this setting, the dataset of optimal agent-environment interactions is explicitly constructed by way of a replay buffer which persists across tasks and is used to train the default policy distribution. This is much more directly in line with standard FTRL, and we can obtain the standard FTRL bound.

**Proposition G.2** (Persistent Replay FTRL Regret; (Orabona, 2019), Corollary 7.9). *Let tasks $M_k$ be independently drawn from $\mathsf{P}_{\mathcal{M}}$ at every round, and let them each be associated with a deterministic optimal policy $\pi_k^\star : \mathcal{S} \to \mathcal{A}$. We make the following mild assumptions: i) $\pi_w(a^\star|s) \geq \epsilon > 0 \,\forall s \in \mathcal{S}$, where $a^\star = \pi_k^\star(s)$ and $\epsilon$ is a constant. ii) $\min_\nu \mathsf{KL}[\nu(\cdot), p(\cdot)] = 0$ asymptotically as $\operatorname{Var}[\nu] \to \infty$. Then with $\eta_{k-1} = L\sqrt{k}$, Algorithm 1 guarantees*

$$\frac{1}{K} \sum_{k=1}^K \ell_k(\nu_k) - \frac{1}{K} \sum_{k=1}^K \ell_k(\bar{\nu}_K) \leq (\mathsf{KL}[\bar{\nu}_K, p] + 1) \frac{L}{\sqrt{K}}, \tag{G.9}$$

*where $\bar{\nu}_K = \operatorname{argmin}_{\nu \in \mathsf{N}} \sum_{k=1}^K \ell_k(\nu)$.*

*Proof.* This follows directly from the arguments made in the proof of Proposition G.1. $\square$

As before, this result can be used to obtain a performance bound for the control policy.

**Proposition G.3** (Control Policy Sample Complexity for MDL-C with Persistent Replay). *Under the setting described in Proposition G.2, denote by $T_k$ the number of iterations to reach $\epsilon$-error for $M_k$ in the sense that $\min_{t \leq T_k} \{V^{\pi_k^\star} - V^{(t)}\} \leq \epsilon$. In a finite MDP, from any initial $\theta^{(0)}$, and following gradient ascent, $\mathbb{E}_{M_k \sim \mathcal{P}_{\mathcal{M}}} [T_k]$ satisfies:*

$$\mathbb{E}_{M_k \sim \mathcal{P}_{\mathcal{M}_i}} [T_k] \geq \frac{80|\mathcal{A}|^2|\mathcal{S}|^2}{\epsilon^2(1-\gamma)^6} \mathbb{E}_{M_k \sim \mathcal{P}_{\mathcal{M}_i} s \sim \operatorname{Unif}_{\mathcal{S}}} \left[ \kappa_{\mathcal{A}}^{\alpha_k}(s) \left\| \frac{d_\rho^{\pi_k^\star}}{\mu} \right\|_\infty^2 \right],$$

*where $\alpha_k(s) := \mathbb{E}_{w \sim \nu} d_{\mathrm{TV}}(\pi_k^\star(\cdot|s), \pi_w(\cdot|s)) \leq \sqrt{\frac{1}{2} G(K)}$,*

$$G(K) := \ell_K(\bar{\nu}_K) + \sum_{k=1}^{K-1} (\ell_k(\bar{\nu}_K) - \ell_k(\nu_k)) + (\mathsf{KL}[\bar{\nu}, p] + 1) L\sqrt{K},$$

*$\kappa_{\mathcal{A}}^{\alpha_k}(s) = \frac{2|\mathcal{A}|(1-\alpha(s))}{2|\mathcal{A}|(1-\alpha(s))-1}$, and $\mu$ is a probability measure over $\mathcal{S}$ such that $\mu(s) > 0 \,\forall s \in \mathcal{S}$.*

*Proof.* Without loss of generality, we select a single state $s \in \mathcal{S}$, observing that the same analysis applies $\forall s \in \mathcal{S}$. For simplicity, we denote $\pi(\cdot|s)$ by $\pi$. We start by multiplying each side of Eq. (G.2)

by $K$ and rearranging:

$$\sum_{k=1}^{K} \ell_k(\nu_k) - \sum_{k=1}^{K} \ell_k(\bar{\nu}_K) \leq (\mathsf{KL}[\bar{\nu}, p] + 1) L\sqrt{K}$$

$$\Rightarrow \ell_K(\nu_K) \leq \sum_{k=1}^{K} \ell_k(\bar{\nu}_K) - \sum_{k=1}^{K-1} \ell_k(\nu_k) + (\mathsf{KL}[\bar{\nu}, p] + 1) L\sqrt{K} \qquad \text{(G.10)}$$

$$= \underbrace{\ell_K(\bar{\nu}_K) + \sum_{k=1}^{K-1} (\ell_k(\bar{\nu}_K) - \ell_k(\nu_k)) + (\mathsf{KL}[\bar{\nu}, p] + 1) L\sqrt{K}}_{:=G(K)}$$

We can multiply both sides by $1/2$ and expand $\ell_K(\nu_K)$:

$$\frac{1}{2} G(K) \geq \mathbb{E}_{w \sim \nu_K} \frac{1}{2} \mathsf{KL}[\pi_K^\star, \pi_w]$$

$$\stackrel{(i)}{=} \mathrm{Var}_{\nu_K} \left[ \sqrt{\frac{1}{2} \mathsf{KL}[\pi_K^\star, \pi_w]} \right] + \mathbb{E}_{\nu_K} \left[ \sqrt{\frac{1}{2} \mathsf{KL}[\pi_K^\star, \pi_w]} \right]^2$$

$$\stackrel{(ii)}{\geq} \left( \mathbb{E}_{\nu_K} \left[ \sqrt{\frac{1}{2} \mathsf{KL}[\pi_K^\star, \pi_w]} \right] \right)^2 \qquad \text{(G.11)}$$

$$\stackrel{(iii)}{\geq} \left( \mathbb{E}_{\nu_K} d_{\mathrm{TV}}(\pi_K^\star, \pi_w) \right)^2$$

where $(i)$ follows from the definition of the variance, $(ii)$ follows from its non-negativity, and $(iii)$ follows from Pinsker's inequality. We then have

$$\mathbb{E}_{\nu_K} d_{\mathrm{TV}}(\pi_K^\star, \pi_w) \leq \sqrt{\frac{1}{2} G(K)}. \qquad \text{(G.12)}$$

Letting $\alpha_K(s) = \sqrt{\frac{1}{2} G(K)}$ and applying Moskovitz et al. (2022a), Lemma 5.2 gives the desired result. $\qquad \square$

## G.2 COMMENT ON IMPROVEMENT ACROSS TASKS

To gain intuition for these bounds, there are several important values of $\alpha(s)$ that we can consider. First, as $\alpha(s) \to 1 - 1/|\mathcal{A}|$, which is the TV distance between a uniform default policy and a deterministic optimal policy, $\kappa_\mathcal{A}^\alpha(s) \to 2$. This is an important value because it's the coefficient obtained for log-barrier regularization—that is, when the default policy is uniform and encodes no information about the task distribution. Next, as $\alpha(s) \to 0$ (that is, as the TV distance between the default policy and the optimal policy decreases), $\kappa_\mathcal{A}^\alpha(s) \to 2|\mathcal{A}|/(2|\mathcal{A}| - 1) < 2$ for $|\mathcal{A}| > 1$. So, we get faster as the distance between the default policy and the optimal policy decreases, as we would hope. Another crucial point to note is that as $|\mathcal{A}| \to \infty$ in this case, $\kappa_\mathcal{A}^\alpha(s) \to 1$. Finally, and importantly for MDL-C, as $\alpha(s) \to 1 - 1/2|\mathcal{A}|$ from below, $\kappa_\mathcal{A}^\alpha(s) \to \infty$. In other words, a sufficiently bad default policy can preclude convergence entirely if it puts too much mass on a suboptimal action. For an illustration of this phenomenon, see Moskovitz et al. (2022a) Figure 4.1. Indeed, this is why our Proposition 4.1 is so useful–by effectively placing an upper bound on $\alpha(s)$ which shrinks as the number of tasks $K$ increases, MDL-C's default policy is guaranteed to a) avoid putting too much mass on a suboptimal action and thereby preclude or delay convergence for the control policy, and b) improve the rate as the default policy regret drops.

## G.3 PARALLEL TASK SETTING

An overview of MDL-C as applied in the parallel task setting is presented in Algorithm 3. One important feature to note is the return threshold $R^\star$. As a proxy for the control policy converging to $\pi_k^\star$, data are only added to the default policy replay buffer when a trajectory return is above this threshold performance (on DM control suite tasks, $R^\star$ corresponded to a test reward of at least 700).

---

**Algorithm 3: Off-Policy MDL-C for Parallel Multitask Learning**

---

1: require: task distribution $\mathsf{P}_{\mathcal{M}}$, policy class $\Pi$
2: initialize: default policy distribution $\nu_1 \in \mathsf{N}$, control replay $\mathcal{D}_0 \leftarrow \emptyset$, default replay $\mathcal{D}_0^\phi \leftarrow \emptyset$
3: initialize control policy parameters $\theta$ and default policy distribution parameters $\phi$.
4: **while** not done **do**
5:     **for** episodes $k = 1, 2, \ldots, K$ **do**
6:         Sample a task $M_k \sim \mathcal{P}_{\mathcal{M}}(\cdot)$ with goal ID feature $g_k$
7:         Collect trajectory $\tau = (\tilde{s}_0, a_0, r_0, \ldots, \tilde{s}_{H-1}, a_{H-1}, r_{H-1}) \sim \mathsf{P}^{\pi_\theta}(\cdot)$, store experience

$$\mathcal{D}_k \leftarrow \mathcal{D}_{k-1} \cup \big\{(\tilde{s}_h, a_h, r_h, \tilde{s}_{h+1})\big\}_{h=0}^{H-1} \tag{G.13}$$

        where $\tilde{s}_h := (s_h, g_k)$.
8:         **if** $R(\tau) \geq R^\star$ (i.e., $\pi_\theta \approx \pi_k^\star$) **then**
9:             Add to default policy replay:

$$\mathcal{D}_k^\phi \leftarrow \mathcal{D}_{k-1}^\phi \cup \big\{(\tilde{s}_h, \pi_\theta(\cdot|\tilde{s}_h))\big\}_{h=0}^{H-1} \tag{G.14}$$

            Note that, e.g., when $\pi_\theta(a|\tilde{s}) = \mathcal{N}(a; \mu(\tilde{s}, g_k), \Sigma(\tilde{s}, g_k))$ is a Gaussian policy, $\mu(\tilde{s}_h, g_k), \Sigma(\tilde{s}_h, g_k)$ are added to the replay with $\tilde{s}_h$.
10:         **end if**
11:     **end for**
12:     Update $Q$-function(s) as in Haarnoja et al. (2018).
13:     Update control policy:

$$\theta \leftarrow \underset{\theta'}{\operatorname{argmin}} \, \mathbb{E}_{\mathrm{Unif}_{\mathcal{D}_k}} \big[ V^{\pi_{\theta'}} - \alpha \mathbb{E}_{w \sim \nu_\phi} \mathsf{KL}[\pi_{\theta'}(\cdot|\tilde{s}_h), \pi_w(\cdot|\tilde{s}_h)] \big] \tag{G.15}$$

14:     Update default policy distribution:

$$\phi \leftarrow \underset{\phi'}{\operatorname{argmin}} \, \mathsf{KL}[\nu_{\phi'}(\cdot), p(\cdot)] + \mathbb{E}_{\mathrm{Unif}_{\mathcal{D}_k^\phi}} \mathbb{E}_{w \sim \nu} \mathsf{KL}[\pi_\theta(\cdot|\tilde{s}_h), \pi_w(\cdot|\tilde{s}_h)] \tag{G.16}$$

15: **end while**

---

We leave more in-depth theoretical analysis of this setting to future work, but note that as the task experience is interleaved, $\bar{\pi}_w = \mathbb{E}_\nu \pi_w$ will converge to the prior-weighted KL barycenter. If, in expectation, this distribution is a TV distance of less than $1 - 1/|\mathcal{A}|$ from $\pi_k^\star$, then the control policy will converge faster than for log-barrier regularization (Moskovitz et al., 2022a).

## H    ADDITIONAL EXPERIMENTAL DETAILS

Below, we describe experimental details for the two environment domains in the paper.

### H.1    FOURROOMS

As input, the agent receives a 16-dimensional vector containing the index of the current state, a flattened $3 \times 3$ local view of its surrounding environment, its previous action taken encoded as a 4-dimensional one-hot vector, the reward on the previous timestep, and a feature indicating the goal state index. The base learning algorithm in all cases is advantage actor critic (A2C; (Mnih et al., 2016)).

**Environment**    The FOURROOMS experiments are set in an $11 \times 11$ gridworld. The actions available to the agent are the four cardinal directions, up, down, left, and right, and transitions are deterministic. In both FOURROOMS experiments, the agent can begin an episode anywhere in the environment (sampled uniformly at random), and a single location with reward $r = 50$ is sampled at the beginning of each episode from a set of possible goal states which varies depending on the experiment and the current phase. A reward of $r = -1$ is given if the agent contacts the walls. All other states give a reward of zero. Episodes end when either a time (number of timesteps) limit is reached or the agent reaches the goal state. Observations were 16-dimensional vectors consisting of

the current state index (1d), flattened $3 \times 3$ local window surrounding the agent (includes walls, but not goals), a one-hot encoding of the action on the previous timestep (4d), the reward on the previous timestep (1d), and the state index of the current goal (1d). In the "goal generalization" experiment, goals may be sampled anywhere in either the top left or bottom right rooms in the first phase and either the top right or bottom left rooms in the second phase. Each phase comprises 20,000 episodes, and in each phase, the agent may start each episode anywhere in the environment. In the first phase, the agent was allowed 100 steps per episode, and in the second phase 25 steps. In the "contingency change" experiment, the possible reward states in each phase were the top left state and bottom right state. In the second phase of training, however, the semantics of the goal feature change from indicating the location of the reward to the location where it is absent. Each phase consisted of 8,000 episodes with maximum length 100 timesteps. Results are averaged over 10 random seeds.

**Agents**   All agents were trained on-policy with advantage actor-critic (Mnih et al., 2016). The architecture was a single-layer LSTM (Hochreiter and Schmidhuber, 1997) with 128 hidden units. To produce the feature sensitivity plots in Fig. 5.1c, a gating function was added to the input layer of the network:

$$x_h = \sigma(b\kappa) \odot o_h, \tag{H.1}$$

where $o_h$ is the current observation, $\sigma(\cdot)$ was the sigmoid funcion, $b \in \mathbb{R}$ is a constant (set to $b = 150$ in all experiments), $x_h \in \mathbb{R}^d$ is the filter layer output, and $\kappa \in \mathbb{R}^d$ is a parameter trained using backpropagation. In this way, as $\kappa_d \to \infty$, $\sigma(b\kappa_d) \to 1$, allowing input feature $o_h, d$ through the gate. As $\kappa_d \to -\infty$, the gate is shut. The plots in Fig. 5.1c track $\sigma(b\kappa_d)$ over the course of training. The baseline agent objective functions are as follows:

$$
\begin{aligned}
\mathcal{J}^{\mathrm{PO}}(\theta) &= V^{\pi_\theta} + \alpha\mathbb{E}_{s \sim d^{\pi_\theta}}\mathsf{H}[\pi_\theta(\cdot|s)] \\
\mathcal{J}^{\mathrm{RPO}}(\theta,\phi) &= V^{\pi_\theta} - \alpha\mathbb{E}_{s \sim d^{\pi_\theta}}\mathsf{KL}[\pi_\theta(\cdot|s), \pi_\phi(\cdot|s)] \\
\mathcal{J}^{\mathrm{VDO-PO}}(\theta) &= \mathbb{E}_{w \sim \nu_\theta}V^{\pi_w} - \beta\mathsf{KL}[\nu_\theta(\cdot), p(\cdot)] \\
\mathcal{J}^{\mathrm{ManualIA}}(\theta,\phi) &= V^{\pi_\theta} - \alpha\mathbb{E}_{s \sim d^{\pi_\theta}}\mathsf{KL}[\pi_\theta(\cdot|s), \pi_\phi(\cdot|s_d)]; \quad s_d = s \setminus g.
\end{aligned}
\tag{H.2}
$$

In all cases $\alpha = 0.1$, $\beta = 1.0$, and learning rates for all agents were set to 0.0007. Agents were optimized with Adam (Kingma and Ba, 2014). Agent control policies were reset after phase 1.

## H.2   DeepMind Control Suite

**Environments/Task Settings**   We use the `walker` and `cartpole` environments from the Deep-Mind Control Suite (Tassa et al., 2018). We consider two multitask settings: sequential tasks and parallel tasks. All results are averaged over 10 random seeds, and agents are trained for 500k timesteps. In the sequential task setting, tasks are sampled one at a time without replacement and solved by the agent. The control policy is reset after each task, but the default policy is preserved. For methods which have a default policy which can be preserved, performance on task $k$ is averaged over runs with all possible previous tasks in all possible orders. For example, when `walker-run` is the third task, performance is averaged over previous tasks being `stand` then `walk` and `walk` then `stand`. In the parallel task setting, a different task is sampled randomly at the start of each episode, and a one-hot task ID vector is appended to the state observation. Learning was done directly from states, not from pixels.

**Agents**   The base agent in all cases was SAC with automatic temperature tuning, following Haarnoja et al. (2018). Standard SAC seeks to optimize the maximum-entropy RL objective:

$$\mathcal{J}^{\mathrm{max-ent}}(\pi) = V^\pi + \alpha\mathbb{E}_{s \sim d^\pi}\mathsf{H}[\pi(\cdot|s)] = V^\pi + \alpha\mathbb{E}_{s \sim d^\pi}\mathsf{KL}[\pi(\cdot|s), \mathrm{Unif}_\mathcal{A}] \tag{H.3}$$

Effectively, then, SAC uses a uniform default policy. The RPO algorithms with learned default policies replace $\mathsf{KL}[\pi(\cdot|s), \mathrm{Unif}_\mathcal{A}]$ with $\mathsf{KL}[\pi(\cdot|s), \pi_w(\cdot|s)]$ (or $\mathsf{KL}[\pi_w(\cdot|s), \pi(\cdot|s)]$). As MDL-C, RPO, and TVPO require that the control policy approximate the optimal policy before being used to generated the a learning signal for the default policy, in the sequential setting, the default policy is updated only after halfway through training. Because variational dropout can cause the network to over-sparsify (and not learn the learn adequately) if turned on too early in training, we follow the strategy of Molchanov et al. (2017), linearly ramping up a coefficient $\beta$ on the variational dropout KL from 0 to 1 starting from 70% through training to 80% through training. Note that MANUALIA is not

applicable to the sequential task setting, as there is no explicit goal feature. In the sequential task setting, we took inspiration from Haarnoja et al. (2018) and Abdolmaleki et al. (2018) and reframed the soft KL penalty for methods with learned default policies as a constraint, i.e.,

$$\max_\pi V^\pi - \alpha \mathbb{E} \mathsf{KL}[\pi_w, \pi] \quad \longrightarrow \quad \max_\pi V^\pi \ s.t. \ \mathbb{E}\mathsf{KL}[\pi_w, \pi] \leq \varepsilon,$$

where $\varepsilon > 0$ was a target KL divergence. Under this formulation, $\alpha$ is treated as a dual variable via Lagrangian relaxation and optimized with the following objective:

$$\max_{\alpha \geq 0} J(\alpha) := \mathbb{E}\alpha \mathsf{KL}[\pi_w, \pi] - \alpha\varepsilon.$$

In the parallel task setting, we convert the base SAC agent into the "multitask" variant used by Yu et al. (2019), in which the agent learns a vector of temperature parameters $[\alpha_1, \ldots, \alpha_K]$, one for each task. In this setting, we found it more effective to set $\alpha$ to a constant value. Test performance was computed by averaging performance across all $K$ tasks presented to the agent. The baseline agent objectives are as in Eq. (H.2), and the Distral objective is given by

$$\mathcal{J}^{\mathrm{Distral}}(\theta, \phi) = V^{\pi_\theta} - \alpha \mathbb{E}_{s \sim d^{\pi_\theta}} \mathsf{KL}[\pi_\theta(\cdot|s), \pi_\phi(\cdot|s)] + \lambda \mathbb{E}_{s \sim d^{\pi_\theta}} \mathsf{H}[\pi_\theta(\cdot|s)].$$

TVPO is trained in the same way as RPO, with the difference being that the default policy objective is to predict the control policy action, rather than a distillation objective. Hyperparameters shared by all agents can be viewed in Table 2.

As a note on performance, Distral performs very strongly in the parallel task setting, with overall performance slightly worse than MDL-C on Walker and virtually the same on Cartpole. However, the gap is significantly greater in the sequential setting, particularly on Walker. We hypothesize that this is due to the fact that by regularizing the control policy to be close to the default policy, but also encouraging the control policy to have high entropy (rather than regularizing the default policy as MDL-C does), Distral can in effect provide a conflicting objective to the control policy when strong structure is present. In particular, on Walker, the optimal policies for each task have significant overlap, and so by encouraging high entropy in the control policy even on the third task, Distral negates the effect of an informative default policy. As evidence, both RPO and TVPO, which only regularize the control policy to be close to the default policy, perform significantly more strongly on Walker in the sequential setting.

| Hyperparameter | Value |
|---|---:|
| Collection Steps | 1000 |
| Random Action Steps | 10000 |
| Network Hidden Layers | 256:256 |
| Learning Rate | $3 \times 10^{-4}$ |
| Optimizer | Adam |
| Replay Buffer Size | $1 \times 10^6$ |
| Action Limit | $[-1, 1]$ |
| Exponential Moving Avg. Parameters | $5 \times 10^{-3}$ |
| (Critic Update:Environment Step) Ratio | 1 |
| (Policy Update:Environment Step) Ratio | 1 |
| Expected KL/Entropy Target | $0.2/-\dim(\mathcal{A})^*$ |
| Policy Log-Variance Limits | $[-20, 2]$ |

Table 2: DM control suite hyperparameters, used for all experiments. *In the parallel setting, $\alpha$ was simply set to 0.1 for methods with learned default policies.

# I   ADDITIONAL EXPERIMENTAL RESULTS

## I.1   FOURROOMS

## I.2   DEEPMIND CONTROL SUITE

| Method | Goal Change | Contingency Change |
|---|---|---|
| PO | $1.25e5 \pm 1.76e4$ | $8.80e4 \pm 1.64e4$ |
| RPO | $1.77e5 \pm 1.11e4$ | $1.04e5 \pm 2.20e4$ |
| VDO-PO | $1.48e5 \pm 1.91e4$ | $8.23e4 \pm 1.98e4$ |
| ManualIA | $1.23e5 \pm 2.51e4$ | $7.69e4 \pm 2.89e4$ |
| MDL-C | $1.08e5 \pm 2.44e4$ | $5.11e4 \pm 1.70e4$ |

Table 3: FourRooms: Average cumulative regret across 8 random seeds in phase 2 of the goal change and contingency change experiments for each method. $\pm$ values are standard error.

| Method | Cartpole | Walker |
|---|---|---|
| SAC | $1.25e5 \pm 1.76e$ | $3.42e5 \pm 6.10e4$ |
| RPO-SAC ($k = 3$) | $1.77e5 \pm 1.11e4$ | $1.04e5 \pm 2.20e4$ |
| VDO-SAC | $1.48e5 \pm 1.91e4$ | $8.23e4 \pm 1.98e4$ |
| MDL-C ($k = 1$) | $1.23e5 \pm 2.51e4$ | $7.69e4 \pm 2.89e4$ |
| MDL-C ($k = 2$) | $1.08e5 \pm 2.44e4$ | $5.11e4 \pm 1.70e4$ |
| MDL-C ($k = 3$) | $1.08e5 \pm 2.44e4$ | $5.11e4 \pm 1.70e4$ |

Table 4: DM Control Suite, Sequential: Average cumulative regret across 8 random seeds in the sequential setting. $\pm$ values are standard error.

| Method | Cartpole | Walker |
|---|---|---|
| SAC | $1.01e5 \pm 2.01e3$ | $1.46e5 \pm 5.11e3$ |
| ManualIA | $9.90e4 \pm 1.87e3$ | $1.50e5 \pm 3.86e3$ |
| MDL-C | $9.47e4 \pm 8.36e2$ | $1.31e5 \pm 1.35e3$ |

Table 5: DM Control Suite, Parallel: Average cumulative regret across 8 random seeds in the parallel task setting. $\pm$ values are standard error.

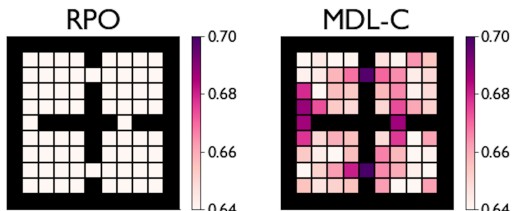

Figure I.1: Heatmaps of $\mathsf{KL}[\pi_\theta(\cdot|s), \pi_w(\cdot|s)] \ \forall s \in \mathcal{S}$ for RPO and $\mathsf{KL}[\pi_\theta(\cdot|s), \pi_{\bar{w}}(\cdot|s)] \ \forall s \in \mathcal{S}$, where $\bar{w} = \mathbb{E}_\nu w$ for MDL-C, averaged over all possible goal states. The RPO default policy nearly perfectly matches the control policy, while the MDL-C default policy diverges most strongly from the control policy at the doorways. This is because the direction chosen by the policy in the doorways is highly goal-dependent. Because the MDL-C default policy learns to ignore the goal feature, it's roughly uniform in the doorways, whereas the control policy is highly deterministic, having access to the goal feature.

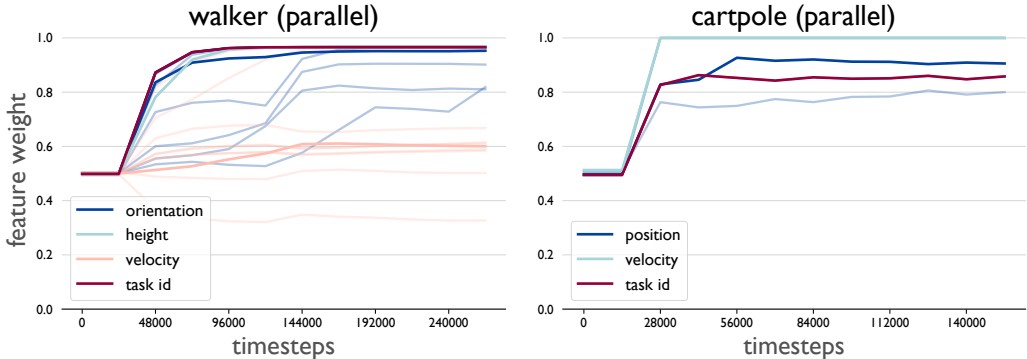

Figure I.2: Without a sparse prior, RPO does not learn to ignore spurious input features.

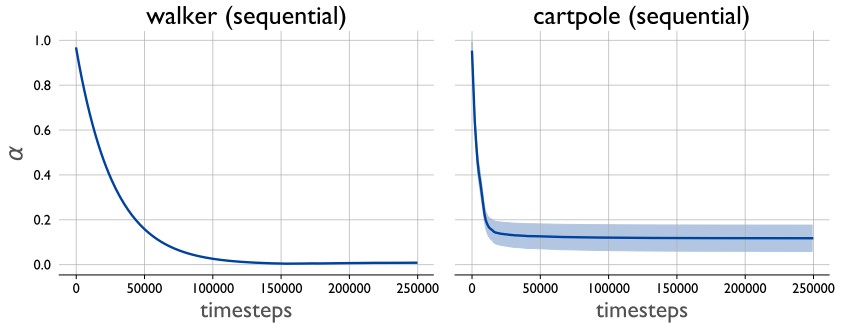

Figure I.3: MDL-C's learned $\alpha$s in the DMC sequential setting. Because $\alpha$ tends to decay, the control policy is able to specialize to the current task later in training. Results averaged over eight random seeds; error shading denotes standard error.

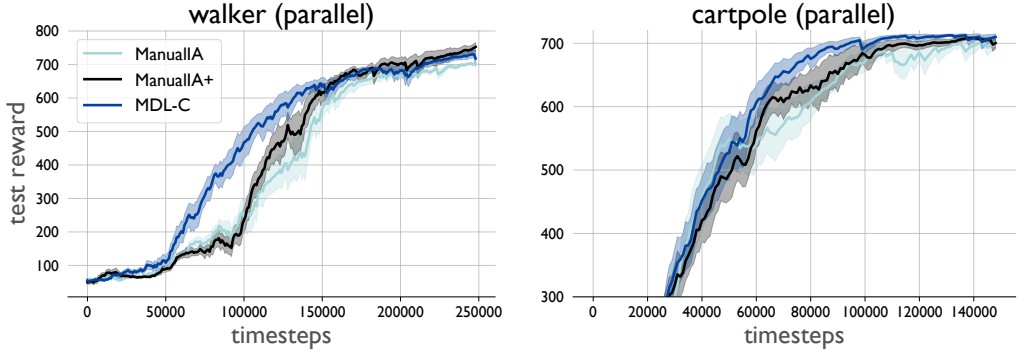

Figure I.4: To test the effect of information asymmetry on its on performance, we trained a variant of MANUALIA in which we withheld the input features that MDL-C learned to gate out (Fig. 5.2) in addition to the task ID feature. We call this modified method MANUALIA+. Average performance is plotted above over 10 seeds, with the shading representing one unit of standard error. We can see that while MANUALIA+ narrowly outperforms MANUALIA, the performance gains of MDL-C can't solely be ascribed to effective information asymmetry.

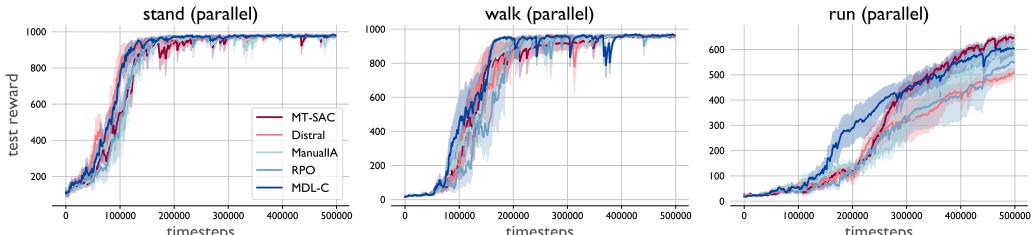

Figure I.5: Test reward on each individual task in the `walker` domain over the course of parallel task training. Average performance is plotted above over 10 seeds, with the shading representing one unit of standard error. We can see the biggest performance difference on `walker, run`, the most challenging task.

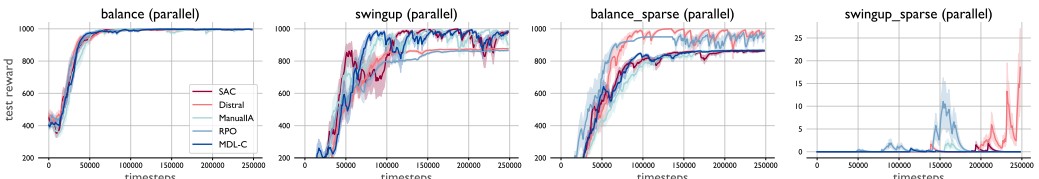

Figure I.6: Test reward on each individual task in the `cartpole` domain over the course of parallel task training. Average performance is plotted above over 10 seeds, with the shading representing one unit of standard error. Interestingly, unlike in the sequential learning setting, joint training seems to impede performance on `swingup_sparse`, with no method succeeding.

