# OpenReview forum: "Minimum Description Length Control"
_ICLR.cc/2023/Conference — ICLR 2023 poster_

### Official Review · Reviewer_rmSg · 2022-10-25

**Confidence:** 3
**Correctness:** 4
**Technical Novelty And Significance:** 3
**Empirical Novelty And Significance:** 2
**Recommendation:** 6

**Clarity, Quality, Novelty And Reproducibility:**

## Clarity, Quality
The paper is overall clear and the writing quality seems good.

## Novelty
The paper provides an interesting perspective.

## Reproducibility
The submission includes codes and the paper has detailed sections for the experiments.

**Strength And Weaknesses:**

# Strength
* The usage of the sparsity induced prior is nicely motivated throughout the paper, the intuition seems quite natural under the paper's constructed scenarios.
* The paper contains both theory and practical results.
* The overall flow of the paper is good and well organized.

# Weakness
* The paper's theoretical study seems a little underwhelming given that it's only in the tabular case, and thus the results do not seem super exciting (more on this below). For example, some recent theory paper already studies multi-task RL in the linear case [1,2,3].
* The literature review does not seem complete, some previous works in multi-task RL are missing, for example, [4]
* Is there any reason that the paper does not include the deep multi-task RL baselines in the experiments? The current baselines do not seem super strong, especially if we want to compare in the dm control suite benchmark.

# Questions:
* Despite the analysis is conducted in the tabular case, the $(1-\gamma)^{-6}$ rate looks quite suboptimal. Intuitively this rate should have nothing to do with the multi-task setting itself. It would be great to have some clarification.
* I am rather confused about the rate w.r.t. the number of task $K$. I may overlook something but is the current result saying that $\kappa^{\alpha^k}$ stays rather constant despite $K$? Thus what's the benefit here in terms of an increasing number of tasks?

[1] Pacchiano, A., Nachum, O., Tripuraneni, N. and Bartlett, P., 2022. Joint Representation Training in Sequential Tasks with Shared Structure.

[2] A. Agarwal, Y. Song, W. Sun, K. Wang, M. Wang, and X. Zhang. Provable benefits of representational transfer in reinforcement learning.

[3] Y. Cheng, S. Feng, J. Yang, H. Zhang, and Y. Liang. Provable benefit of multitask representation learning in reinforcement learning.

[4] Emma Brunskill and Lihong Li. Sample complexity of multi-task reinforcement learning.

**Summary Of The Paper:**

The paper introduces a new perspective for multi-task RL that to distill each control policies in each task using techniques inspired from the minimum description length principle. Specifically, the proposed algorithm learns a policy parameters distribution that both contains the control policy parameters and constraint to be close to the sparsity induced prior. The paper then shows some motivation of the minimum description length control with some constructed scenarios and show the performance analysis in the tabular case. Finally, the paper has some experiments on both synthetic and popular RL benchmarks.

**Summary Of The Review:**

Overall the idea of enforcing simpler representations into multi-task RL is quite interesting. However, the theory result is a little underwhelming, and the experiments may need stronger baselines. Also there is still some confusion around the current version. Thus I would recommend a weak reject at this point.

---

> ### Author Response · Authors · 2022-11-15
> **Author Response Part 1/2**
>
> Thank you very much for your careful review and suggestions! We believe they have certainly helped us to improve the paper. We are also glad that you find MDL-C to be an interesting idea and the paper well-organized and clear. We aim to address your points below:
>
> - Theoretical Study: Thank you for the links to these interesting papers! We agree that it would be very interesting to study MDL-C using a linear parameterization, and in fact have already cited [1]. However, [1, 2, 3] all rely on a different type of assumption regarding the form of structure shared among the MDPs with which the agent is faced. Specifically, the core assumption made by this strain of work is that the transition dynamics are linearly decodable from a set of features which is shared across tasks or in which the transition matrix admits a low-rank decomposition. This is very different from, and in our opinion, more restrictive than, our structural assumption—that is, in its simplest form, that the optimal policies of the tasks with which our agents are faced take similar actions in at least some part of the state space. Beyond this, the MDPs in $\mathcal M$ need only share the same state and action space, with no direct assumptions about transitions or rewards.  This is important, because the assumed structures in the transition distribution made by [1, 2, 3] act as the starting points for algorithm development. MDL-C/RPO/TVPO however, can leverage similarity among optimal policies when it exists, but are not dependent on it as a prerequisite. (E.g., TVPO (and RPO/MDL-C) is guaranteed to perform no worse than log-barrier regularization, which has a polynomial sample complexity guarantee.) We chose a tabular parameterization with no further assumptions because it bore the greatest similarity to MDL-C’s use in practice while still admitting analysis. MDL-C is not designed to leverage structure in the transitions (and thus to use the linear structure assumed in [1, 2, 3] would need to be a different algorithm), but we agree it would be a very interesting future research direction to add this ability as well. Ideally, we’d like a generalist method which can identify on its own and exploit different types of structure in the environment. We have added this discussion to Appendix B.
>
> - Literature Review: Thank you very much for sharing this very interesting paper with us! We have added references in Section 2 and Appendix C.
>
> - Additional Baselines: We have added experiments with Distral (Teh et al. ‘17) and RPO to the parallel control suite experiments and Distral and TVPO (Moskovitz et al. ‘22) to the sequential control suite experiments, as well as discussion and analysis of these results. Note that TVPO is very similar to RPO, with the main difference being that the default policy is trained to predict the control policy’s actions rather than distill its distribution. As a brief summary, Distral performs very strongly in the parallel task setting, with overall performance slightly worse than MDL-C on Walker and virtually the same on Cartpole. However, the gap is significantly greater in the sequential setting, particularly on Walker. We hypothesize that this is due to the fact that by regularizing the control policy to be close to the default policy, but also encouraging the control policy to have high entropy (rather than regularizing the default policy as MDL-C does), Distral can in effect provide a conflicting objective to the control policy when strong structure is present. In particular, on Walker, the optimal policies for each task have significant overlap, and so by encouraging high entropy in the control policy even on the third task, Distral negates the effect of an informative default policy. As evidence, both RPO and TVPO, which only regularize the control policy to be close to the default policy, perform significantly more strongly on Walker in the sequential setting. This effect does not come into play as much in the parallel setting, as all tasks are presented together. We have added this discussion to Appendix H.2.

---

> > ### Author Response · Authors · 2022-11-15
> > **Author Response Part 2/2**
> >
> > Answers to questions:
> >
> > - Convergence Rate: Indeed, this rate is not particularly fast! However, to our knowledge, there has not been a demonstrated faster rate wrt the discount factor for policy gradient methods with no further assumptions on the MDP/task using a tabular softmax parameterization (see Agarwal et al. ‘20  for further details). The key thing is to guarantee a polynomial convergence rate (and avoid an exponential one). As for its relationship to the multi-task setting, one can observe that $(1 - \gamma)^{-6}$ lies outside the expectation taken wrt the task distribution, and thus this component of the bound is, as you note, independent of the multi-task setting itself.
> >
> > - Benefit of More Tasks: This is a very interesting and subtle point. There are several important values of $\alpha(s)$ that we can consider. First, as $\alpha(s) \to 1 - 1/|\mathcal A|$, which is the TV distance between a uniform default policy and a deterministic optimal policy, $\kappa_\mathcal A^\alpha(s) \to 2$. This is an important value because it’s the coefficient obtained for log-barrier regularization—that is, when the default policy is uniform and encodes no information about the task distribution. Next, as $\alpha(s) \to 0$ (that is, as the TV distance between the default policy and the optimal policy decreases), $\kappa_\mathcal A^\alpha(s) \to 2|\mathcal A| / (2|\mathcal A| - 1) < 2$ for $|\mathcal A| > 1$. So, we get faster as the distance between the default policy and the optimal policy decreases, as we would hope. Another crucial point to note is that as $|\mathcal A|\to \infty$ in this case, $\kappa_\mathcal A^\alpha(s) \to 1$.  Finally, and importantly for MDL-C, as $\alpha(s) \to 1 - 1/2|\mathcal A|$ from below, $\kappa_\mathcal A^\alpha(s) \to \infty$. In other words, a sufficiently bad default policy can preclude convergence entirely if it puts too much mass on a suboptimal action. For an illustration of this phenomenon, see Moskovitz et al. ‘22 Figure 4.1. Indeed, this is why our Proposition 4.1 is so useful–by effectively placing an upper bound on $\alpha(s)$ which shrinks as the number of tasks $K$ increases, MDL-C’s default policy is guaranteed to a) avoid putting too much mass on a suboptimal action and thereby preclude or delay convergence for the control policy, and b) improve the rate as the default policy regret drops. We have added this discussion to Appendix G.2.
> >
> > We sincerely thank you once again for helping us to strengthen our paper and for your insights and suggestions. If you believe these modifications constitute an improvement to the paper, we hope you might consider raising your score, and if not, we’d be more than happy to make additional changes and/or engage in further discussion.

---

> > > ### Comment · Reviewer_rmSg · 2022-11-19
> > > **Response**
> > >
> > > I agree the new results + the clarification + better compassion and reviews of other multi-task papers indeed make the paper stronger and more complete. Thus I flipped my evaluation after the rebuttal. I appreciate the authors for their great efforts and the construction conversation!

---

### Official Review · Reviewer_uHBy · 2022-10-28

**Confidence:** 2
**Correctness:** 4
**Technical Novelty And Significance:** 3
**Empirical Novelty And Significance:** 2
**Recommendation:** 8

**Clarity, Quality, Novelty And Reproducibility:**


### Clarity
This paper is written well, in clear language. In the experiments section, the paragraphs for each setting could be split up for clarity.

### Quality
I believe that the research here is of good quality.

### Novelty
I haven't seen work like this elsewhere, to the best of my knowledge.

### Reproducibility
Code and experimental details provided.

**Strength And Weaknesses:**

### Strengths
- Good structuring of the exposition.
- Clear intuitive explanation before jumping into the maths
- Nice theoretical analysis
- Experiments seem good, with both simple and complex tasks.

### Weaknesses
- Some paragraphs could be split up.
-

**Summary Of The Paper:**

This paper provides a method of combining the minimum description length principle to the finding of optimal policies for multitask RL problems. This allows the model to trade off between adapting to new information, and maintaining simplicity, thus adapting to epistemic uncertainty naturally.

After introducing the method, the paper provides discussions about interpretations of some key terms, and carries out some theoretical analyses regarding performance and sample complexity.

Experiments are carried out in the FOURROOMS environment and the DeepMind Control suite against several baselines.

**Summary Of The Review:**

My score is based on the fact the seeming novelty of the work, how well it is written, as well as the suite of experiments carried out. I wasn't able to check the correctness of the theory.

---

> ### Author Response · Authors · 2022-11-15
> **Author Response**
>
> Thank you very much for your review! We’re glad that you liked the paper. We had to condense some paragraphs due to space constraints, but we will certainly do our best to split up the longer paragraphs in the camera ready version if the paper is fortunate enough to be accepted.

---

### Official Review · Reviewer_TuJe · 2022-10-28

**Confidence:** 3
**Correctness:** 4
**Technical Novelty And Significance:** 3
**Empirical Novelty And Significance:** 3
**Recommendation:** 5

**Clarity, Quality, Novelty And Reproducibility:**

Clarity:

The clarity can be improved (see weaknesses).

Quality:

The quality of the paper is good since the authors provide a novel algorithm, some theoretical results, and experiments.


Novelty:

The algorithm is novel, although is not clear what are the main "philosophical" differences with respect to

Reproducibility:

The authors provide code to reproduce the results. However, I encourage the authors to provide more details in the README to reproduce the results in an easier way (and provide a file with dependencies).




**Strength And Weaknesses:**

Strengths:

- The problem is relevant to the community.

- As far as I know, the idea of using the "minimum description length" principle is novel and well-motivated. The idea is very interesting.


Weaknesses:

- The main weakness of the paper is the clarity. The paper, in general, is well-written, but some parts are hard to be understood and the connection with the SotA is not very clear. For example, what is the main difference between the proposed method and Moskovitz 2022? I am sure that there are differences between the two approaches, but I would like to see this difference written more clearly in the paper.

- The performance analysis section could be improved. For example, this sentence is not very clear:
"is to obtain an upper bound on the average KL between default policies sampled from the default policy distribution and an optimal policy for a task sampled from the default policy distribution"
Maybe is a task sampled from the task distribution?

- Default policy performance section: The connection with FTRL is quite interesting, although here we are not in an adversarial setting, so we are more interested in minimizing the internal regret instead of the external one, or am I missing something?

- Control policy performance section: In proposition 4.2, the $\ge$ is a typo and has to be instead a $\le$? Why do we have this dependency of $(1-\gamma)^6$?

- Experiments: why the authors do not compare their method with the principled RPO algorithm proposed in Moskovitz 2022 (TVPO)?

- The plots are not readable for color-blind people.

**Summary Of The Paper:**

The paper deals with the relevant multi-task RL problem. The authors propose a novel method that follows the "minimum description length" principle, to learn a common structure between the tasks. The aim of the proposed approach is to improve the generalization. The authors provide sample complexity results in the finite state-action spaces and provide some experiments to show the effectiveness of the novel algorithm.

**Summary Of The Review:**

See Weaknesses and Strengths.

---

> ### Author Response · Authors · 2022-11-15
> **Author Response Part 1/2**
>
> Thank you for your detailed review and helpful advice! We're glad that you find MDL-C to be an interesting and well-motivated idea. We aim to address your points below:
>
> - Proposition 4.2: Thank you for this comment, we apologize that the result was unclear! It’s actually not a typo, though the statement of the result could be more clear. It should be understood as “if the agent has experienced at least $T_k$ samples on task $k$, it’s guaranteed to have reached less than $\epsilon$ error by that time.” The factor of $(1 - \gamma)^{-6}$ comes from standard bounds on the performance of softmax tabular policy gradient methods (Agarwal et al. ‘20) and is independent from our algorithm (that is, it’s a dependence shared by any KL-regularized policy gradient method under this parameterization). Specifically, it comes from Agarwal et al. ‘20 Corollary 5.1 and standard convergence results for gradient ascent (restated in Agarwal et al. ‘20 Thm E.1).
>
> - Clarity: We apologize for any lack of clarity! We have taken the following steps to improve clarity in the new draft:
>   - We have modified the language of Proposition 4.2 to clarify the meaning of $T_k$.
>   - We have added additional discussion of baseline methods used in experiments in Appendix H.
>   - We have added additional description of the code and added a requirements.txt file for dependencies to the supplementary material.
>   - We have added additional discussion of related multi-task RL frameworks in Appendix B.
>   - We have added further explanation of the theoretical bounds in Appendix G.2.
>   - We have added an explanation of the connection between MDL-C and TVPO to Appendix C in the paper, which we reproduce below:
>
> Another important method in the sequential setting is TVPO (Moskovitz et al. ‘22), in which (in the tabular case) the default policy is defined as a softmax over the average action frequencies of the optimal policies for the tasks that the agent has seen so far. That is, if the average optimal action in a state $s$ is given by
> $$
>  \hat\xi_k(s,a) = \frac{1}{k}\sum_{i=1}^k 1(\pi_i^\star(s) = a),
> $$
> then the TVPO default policy is
> $$
>  \pi_w(a|s) = \mathrm{softmax}\left(\hat\xi_k(s,a) / \beta(k)\right),
> $$
> where $\beta(k)$ is a temperature which decays as $k\to\infty$. In high-dimensional state and action spaces, this tabular solution can be approximated by training a default policy to predict the converged control policy's actions in each task. Importantly, this is equivalent to using KL distillation in that the default policies will converge to the same barycenter policy Moskovitz et al. ‘22 as long as the distillation is only performed once the control policy has converged in each task. Using KL distillation in this way is exactly the RPO baseline that we use in this paper. Crucially, the use of the softmax with decaying temperature was introduced by Moskovitz et al. ‘22 as a useful `hack' to prevent the default policy from overfitting to early tasks, as the optimal default policy is the barycenter policy (approximated as the number of draws from the task distribution grows). Thus, MDL-C can itself be seen as a scalable advancement of TVPO which models the agent's epistemic uncertainty about the task distribution by placing a sparse prior over the default policy parameters (and uses a distillation loss rather than action prediction). In other words, MDL-C represents a principled approach to reducing the risk of default policy overfitting in the low-data regime.
>
>
> Performance Analysis: Ah, thank you! This was a typo, it should be “task sampled from the task distribution” not sampled from the default policy distribution. We have modified this in the new draft.
>
> FTRL: The key point in this case is that we simply assume that tasks are sampled from $\mathsf P_\mathcal M$, and the agent does not have any knowledge of the details of this sampling process nor does it attempt to model them. The FTRL regret bound we use does not make assumptions regarding the sampling process / task ordering (i.e., if the task sampling process is viewed as an opponent, no assumptions are made about it).

---

> > ### Author Response · Authors · 2022-11-15
> > **Author Response Part 2/2**
> >
> > Experiments: We agree, this is a useful point of comparison. We have added TVPO (Moskovitz et al. ‘22) in addition to Distral (Teh et al. ‘17) as baselines in the control suite experiments, along with accompanying discussion. To summarize, TVPO is essentially equivalent to RPO in the infinite data limit, the main difference being that RPO uses a distributional loss (via KL distillation) to learn the default policy, while TVPO’s default policy explicitly predicts the actions taken by the control policy. As detailed by Moskovitz et al. ‘22, the main shortcoming of previous distillation-based methods was that they updated the default policy using suboptimal experience collected by the control policy before convergence. The default policies for RPO and TVPO converge to the same optimum, as the TV and KL barycenters are the same (Moskovitz et al. ‘22 Lemmas 5.1 and D.1), the difference is only in the loss used to train them (once the algorithm has been modified to only distill the control policy into the default policy once it has converged).
> >
> > Readability: Thank you for pointing this out, and we apologize for the oversight! We have updated the color scheme to be color-blind readable: https://gka.github.io/palettes/#/9|d|00429d,96ffea,ffffe0|ffffe0,ff005e,93003a|1|1
> >
> > Thank you once again for your very helpful comments and suggestions. As a result, we believe that we've been able to strengthen the paper, particularly with regards to clarifying the text, improving the experimental evaluation, and improving accessibility. We hope that these changes have ameliorated your concerns. If so, we would be very appreciative if you might consider raising your score. We are also happy to discuss and address any additional concerns.

---

### Official Review · Reviewer_DtWg · 2022-10-31

**Confidence:** 3
**Correctness:** 3
**Technical Novelty And Significance:** 3
**Empirical Novelty And Significance:** 3
**Recommendation:** 6

**Clarity, Quality, Novelty And Reproducibility:**

$\textbf{Clarity: }$

The writing and presentation of the proposed method is almost clear. The organization of this paper is good.

&nbsp;

$\textbf{Novelty: }$

Although a few prior works present similar high-level ideas, to my knowledge, the theoretical results and the proposed method are novel.

&nbsp;

$\textbf{Quality: }$

The theoretical results are clear and the proposed method is almost clear and sound yet I also have questions for the authors. The empirical evaluation and analysis is kind of insufficient in my personal opinion.

&nbsp;


$\textbf{Reproductibility:}$

The proposed method is clear and most experimental details are provided in the appendix. The source codes are also provided.


**Strength And Weaknesses:**

$\textbf{Strengths:}$
+ The paper is well written and organized.
+ The theoretical results of the proposed algorithm is discussed (for both the control policy and the default policy).
+ The related work and background are well enclosed and connected in the text.
+ Multiple settings and evaluation/analysis aspects are considered in the experiments.

&nbsp;


$\textbf{Weaknesses and Questions: }$



According to Algorithm 3 in the appendix, the default policy replay is only added when the control policy is near-optimal, based on which the default policy distribution is trained. Meanwhile, the control policy has a KL constraint on the default policy distribution. I am wondering how the initial phase is dealt with, in another word, is it possible that the initial default policy distribution prevents the control policy from learning a near-optimal policy and in turn the default policy replay remains empty?

I think this may be related to a proper scheduling of the hyperparameter $\alpha$. Thus, how is $\alpha$ selected or scheduled? And how different choices of the value or schedule influence the performance?


&nbsp;



For the experiments, my first question is on the baseline method RPO. According to the formulation in Equation H.2, assuming $\pi_{\phi}$ to be the default policy, what does ‘regularized policy optimization with no constraint on the default policy (RPO)’ mean? Does it mean that the $\pi_{\phi}$ is learned by MLE (or Behavior cloning), i.e., without adopting the KL with a prior distribution?


&nbsp;


I recommend the authors to additionally consider other domains and tasks in DMC suite (e.g., Finger and Quadruped). Two domains are insufficient to me for a convincing experimental evaluation.

I appreciate that the authors provide the feature weight curves which help a lot in understanding the effectiveness of MDL-C. I am curious about the feature weight curves of the baseline methods (e.g., PO and RPO), can the authors provide the curves especially for DMC? If not, what is the reason?

Another question is, why is RPO not considered in the parallel setting? And similarly, ManualIA is not shown in the sequential setting.


&nbsp;


Finally, as the authors present a motivation from the perspective of generative model of optimal policy parameters, with on a group-based assumption. I think it will be interesting and insightful if the authors analyzes the learned optimal parameters (since a shared policy may be used, it may consider the learned representation of features) a posteriori.


**Summary Of The Paper:**

This paper studies multitask RL and proposes a new framework based on minimum description length (MDL) principle, called MDL-control (MDL-C). The framework contains the learning of the control policy and the default policy. The control policy is trained according to Regularized Polic Optimization regarding the default policy, which is expected to learn the common structure; while the default policy (distribution) is formalized as the hypothesis that meets the MDL principle. Sparsity-inducing priors and variational dropout are motivated and adopted for the effective implementation of learning the default policy distribution. This paper also presents the theories on the regret upper bound of the default policy and the complexity of learning near-optimal control policy. In the experiments, the proposed method is evaluated in FoorRoom and DMC suite with multiple settings, demonstrating the effectiveness in learning desired default policy and the superiority over the considered baseline methods.

**Summary Of The Review:**

According to my detailed review above, I think this paper is marginally above the acceptance threshold mainly due to the novelty of the proposed framework along with the theoretical results.

---

> ### Author Response · Authors · 2022-11-15
> **Author Response**
>
> Thank you for your detailed review and kind words about the paper! We hope to address your concerns and questions below:
>
> - Initial Learning and $\alpha$: Indeed, these are important points. Regarding the default policy distribution precluding the control policy from reaching strong performance–this is in fact one of the primary motivations for MDL-C versus RPO. By regularizing the default policy distribution, there is a much lower chance that this issue occurs. Without regularization, the default policy can overfit to the previous task(s), causing it to mislead and thereby delay or prevent convergence of the control policy. This possibility is discussed in more detail in Moskovitz et al. ‘22. For most experiments, we just used a fixed $\alpha$, but for the DMC experiments, we used the trust region-inspired approach used by SAC (Haarnoja et al. ‘18) and MPO (Abdolmaleki et al. ‘18) and reformulate the regularization penalty as a constraint on the deviation of the control policy from the default policy. In this framework, $alpha$ takes the form of a Lagrange multiplier whose gradient is the agent’s violation of this KL constraint. In practice, this leads to decay of $\alpha$ over the course of training (Fig. I.3), allowing the control policy to specialize to the current task. Further details are described in Appendix Section H.2.
>
> - RPO: Yes, that’s correct–RPO learns the default policy via distillation, but without the sparse prior. In other words, it’s MDL-C but without regularization of the default policy. We have added additional text emphasizing this distinction.
>
> - Additional Domains: We agree that additional domains would be helpful! We are working to add more domains. As the experiments (particularly in the sequential setting) are rather expensive, if runs do not finish before Friday, we will post tables of results as comments instead. Thank you for your understanding!
>
> - Feature Curves: Thank you! This is a great suggestion, and we’ve added the feature curves for RPO on DMC to the appendix (Fig. I.2). As expected, RPO does not learn to gate out task-specific information from the inputs to its default policy.
>
> - RPO in the Parallel Setting: Thank you for pointing this out! We have added RPO to the experiments in the parallel setting. This is a valuable point of comparison, as in this setting RPO will not overfit to “previous” tasks because all tasks are presented together, and differences in performance are likely due to the sparsity (and particularly, suppression of spurious features) induced by the MDL prior placed on the default policy. As seen in the updated Figure 5.2, we can further confirm the importance of this regularization on performance, as not only does MDL-C outperform RPO on both domains, but the difference in performance is greater on Walker, where a higher proportion of input features are ignored by MDL-C’s default policy. As mentioned above, we additionally added plots of the feature weights learned by RPO in Fig. I.2.
>
> - ManualIA in the Sequential Setting: ManualIA relies on a priori knowledge of the feature(s) that should be disregarded by the default policy, typically a feature indicating the current goal/task. In the parallel setting, this is possible because the agent is provided with a feature indicating the current task at the beginning of each episode. In the sequential setting, however, there is no goal/task id feature provided to the agent, so the prior knowledge of what feature(s) to ignore is lacking. This is, in our opinion, a significant advantage of MDL-C with respect to methods which require, in essence, manual feature pruning. We have added comments emphasizing this to the updated paper in Section 5.2.
>
> - Analysis of Learned Parameters: We agree that this would be an interesting analysis! We will add such an investigation to a future draft.
>
> Once again, we thank you for your detailed comments and suggestions, and are glad you liked the paper! We hope that these modifications and responses improve the paper in your view, and we’d be happy to discuss further if this is not the case.

---

### Author Response · Authors · 2022-11-15
**Updated Paper Draft**

We offer a sincere thank you to all reviewers for your insightful and detailed comments and suggestions, and for your patience as we prepared our responses, which you may find below. We have uploaded a new draft of the paper which contains the following general improvements:

- Additional clarification on our method and its relationship to baselines.
- Additional discussion of the obtained theoretical results.
- Additional baselines in the control suite experiments: TVPO and Distral in the sequential setting and RPO and Distral in the parallel setting.
- Additional literature review.
- More details and requirements for running the code have been added to the supplementary material.
- The colors used in the plots are now color-blind friendly.

Descriptions of more specific changes are detailed in our responses below. We thank you all once again for your helpful reviews and look forward to a productive discussion!

---

### Decision · Program_Chairs · 2023-01-20

**Decision:**

Accept: poster

**Justification For Why Not Higher Score:**

We do not have any high-confident reviewer who championed this work.

**Justification For Why Not Lower Score:**

Most reviewer are positive about this work. It is novel and mostly a well-written and well-executed work.

**Metareview: Summary, Strengths And Weaknesses:**

The paper considers the multi-task RL problem and suggests using a KL-regularized policy optimization with a prior that is distilled from solving previous tasks. The distillation procedure is formulated following the Minimum Description Length (MDL) principle. In particular, a two-part MDL is used in which the likelihood is defined over the state-action pairs of the optimal policy of solved tasks, and the hypothesis code-length is defined based on the KL distance w.r.t. a sparsity-inducing prior. The result of this MDL procedure is going to be the new policy prior in solving future tasks.

The reviewers are mostly positive about this work. They believe the paper is well-written, the idea of using MDL is novel and well-motivated, and experiments are reasonable. We had some discussions and I believe the authors adequately answered most questions.

My main concern is regarding the theoretical guarantee of Proposition 4.2. I should be upfront and say that this is mostly a presentation issue and not a fundamental flaw.

That proposition, as stated, provides a lower bound on the expected number of iterations $\mathbf{E}[T_k]$ such that $\min_{t \leq T_k} [ V^{\pi_k} - V^{(t)} ] \leq \epsilon$.
But why should we care about a lower bound without having a matching upper bound? Is there any guarantee that we do not need $|S|^{10}$ times more of the lower bound of Proposition 4.2 before having a small error of epsilon? Note that a lower bound alone does not preclude that possibility. For all that matters, we could say that zero is a lower bound on the expected value of $T_k$. It would be valid, but useless.

This issue was brought up by Reviewer TuJe, who thought the direction of the inequality was a typo. This was not followed up by that reviewer, but I think is important because in its current form, the theoretical result is not meaningful.

I looked at the proof. It relies on Lemma 5.2 of Moskovitz et al. (2022a). That lemma has the same issue in my opinion. Looking at the proof of that lemma, we see that Lemma 4.5 is used, which guarantees that as soon as T is larger than a certain value, the difference between the optimal value function and the obtained value is going to be smaller than epsilon. Lemma 4.5 is clear. Even though it is presented as if it is a lower bound, its statement allows us to obtain the smallest T to reach a small error.

Taking the expectation of both sides of (D.6) of Moskovitz et al. (2022a) makes the interpretation unclear. It does not say that a certain number of steps is enough to reach a small error anymore. It says that in average we need at least these many steps, but leaves the room open to requiring many more steps for the guarantee to hold. That, however, is not the right (and I believe intended) interpretation. The right interpretation is that the number of steps required to reach a certain level of error, averaged over tasks, is sufficient to be a certain value. The result should not be presented as a lower bound.

This is a presentation issue, and I hope the authors can revise their paper to clarify it.

I also want to note that there are some other issue such as having a leftover "Note: In the above, there is a small error", which seems to be an internal comment by co-authors, just before Eq. (G.4) or several uncompiled references in Appendix B.

Overall, based on the recommendation of the reviewers and my own reading of the paper,  the paper's merit outweighs its shortcomings. I recommend its acceptance with the understanding that the authors would consider all reviewers comments, including improving the clarity of the theoretical result.

**Note From Pc:**

if the above contains the word "oral" or "spotlight" please see: "oral" presentation means -> notable-top-5% and "spotlight" means -> notable-top-25%. As stated in our emails, we are disassociating presentation type from AC recommendations